# GREB1 induced by Wnt signaling promotes development of hepatoblastoma by suppressing TGFβ signaling

Shinji Matsumoto [1,7], Taku Yamamichi [1,2,7], Koei Shinzawa [1], Yuuya Kasahara[3,4], Satoshi Nojima[5], Takahiro Kodama [6], Satoshi Obika [3,4], Tetsuo Takehara[6], Eiichi Morii[5], Hiroomi Okuyama[2] & Akira Kikuchi [1]

The *β-catenin* mutation is frequently observed in hepatoblastoma (HB), but the underlying mechanism by which Wnt/β-catenin signaling induces HB tumor formation is unknown. Here we show that expression of growth regulation by estrogen in breast cancer 1 (GREB1) depends on Wnt/β-catenin signaling in HB patients. GREB1 is localized to the nucleus where it binds Smad2/3 in a competitive manner with p300 and inhibits TGFβ signaling, thereby promoting HepG2 HB cell proliferation. Forced expression of β-catenin, YAP, and c-Met induces HB-like mouse liver tumor (BYM mice), with an increase in *GREB1* expression and HB markers. Depletion of GREB1 strongly suppresses marker gene expression and HB-like liver tumorigenesis, and instead enhances TGFβ signaling in BYM mice. Furthermore, antisense oligonucleotides for GREB1 suppress the formation of HepG2 cell-induced tumors and HB-like tumors in vivo. We propose that GREB1 is a target molecule of Wnt/β-catenin signaling and required for HB progression.

---

[1] Departments of Molecular Biology and Biochemistry, Graduate School of Medicine, Osaka University, 2-2 Yamadaoka, Suita 565-0871, Japan. [2] Departments of Pediatric Surgery, Graduate School of Medicine, Osaka University, 2-2 Yamadaoka, Suita 565-0871, Japan. [3] Graduate School of Pharmaceutical Sciences, Osaka University, 1-6 Yamadaoka, Suita, Osaka 565-0871, Japan. [4] National Institutes of Biomedical Innovation, Health and Nutrition (NIBIOHN), 7-6-8 Saito-Asagi, Ibaraki 567-0085, Japan. [5] Departments of Pathology, Graduate School of Medicine, Osaka University, 2-2 Yamadaoka, Suita 565-0871, Japan. [6] Departments of Gastroenterology and Hepatology, Graduate School of Medicine, Osaka University, 2-2 Yamadaoka, Suita 565-0871, Japan. [7] These authors contributed equally: Shinji Matsumoto, Taku Yamamichi. Correspondence and requests for materials should be addressed to A.K. (email: akikuchi@molbiobc.med.osaka-u.ac.jp)

The Wnt/β-catenin pathway plays an important role in cell proliferation and differentiation during development and tissue homeostasis, and its deregulation is associated with numerous diseases[1]. In physiological conditions, Wnt stabilizes β-catenin by inhibiting the Axin function, and stabilized β-catenin is then translocated to the nucleus, where it binds to transcription factors, T-cell factor (TCF) and lymphocyte enhancer factor (LEF), leading to expression of various genes[2]. Mutations in Wnt/β-catenin signaling components occur frequently in cancer and result in constitutive β-catenin accumulation[3]. For example, the *APC* gene is mutated in 70–80% of colorectal cancer cases and the *β-catenin* gene (*CTNNB1*) is mutated in 15–20% of cases. In hepatocellular carcinoma (HCC), the *β-catenin* and *Axin* genes are mutated in around 30% and 5–10% of cases, respectively[4]. Although rates of active mutations of the *β-catenin* gene in adult HCC vary among tumors associated with different etiologies, a high rate (50–90%) of mutations in the *β-catenin* gene was found in hepatoblastoma (HB)[5].

HB is the predominant hepatic neoplasm in infants and young children, with an incidence of a few cases per 1 million children[6]. HB differs from HCC by distinct morphological patterns reminiscent of hepatoblasts and their arrangement in the developing liver[7]. Clinically, advances in surgery and postoperative chemotherapy have improved outcomes for HB, resulting in 5-year survival rates averaging 82%[6]. However, there are still aggressive forms that remain difficult to treat. Therefore, new treatments are needed for advanced-stage tumors, and an understanding of HB pathobiology is necessary for developing targeted therapies.

Growth regulation by estrogen in breast cancer 1 (GREB1) is a gene induced by estrogen in MCF7 breast cancer cells[8], and expressed in estrogen receptor α (ERα)-positive breast cancer cells but not in ERα-negative cells. ERα binds to the promoter regions of the *GREB1* gene, and expresses GREB1, which—in turn— interacts directly with ERα and activates its transcriptional activity[9]. Knockdown and overexpression of GREB1 suppresses and promotes proliferation of breast cancer cells, respectively[10]. The GREB1 promoter region has an androgen response element, GREB1 is induced by androgen in androgen receptor (AR)-positive prostate cancer cells[11]. GREB1 knockdown also inhibits the proliferation of AR-positive prostate cancer cells. Thus, GREB1 could be a potential therapeutic target for hormone-sensitive cancers. However, it remains unclear whether GREB1 expression is involved in tumor formation in cancers that are not hormone-sensitive.

In this study, we identified GREB1 as an uncharacterized target gene expressed by Wnt/β-catenin signaling, and found that GREB1 expression is critical for HB cell proliferation. GREB1 was frequently detected together with β-catenin in the tumor lesions of HB patients, and GREB1 inhibited TGFβ signaling, and thereby promoting HB cell proliferation. In addition, GREB1 depletion inhibited HB cell proliferation in vitro and in vivo. Here we propose a function of GREB1 in HB cells and the possibility of a therapeutic strategy for HB using amido-bridged nucleic acid (AmNA)-modified antisense oligonucleotides (ASOs) that target GREB1.

## Results

**GREB1 is a target gene of Wnt/β-catenin signaling in HB.** To clarify the mechanism of tumorigenesis of HB, we screened uncharacterized downstream target genes of Wnt/β-catenin signaling in HepG2 HB cells, which were established from liver tumors with characteristics of HB and had a truncated mutation of the *β-catenin* gene at exons 3 and 4[5,12]. RNA-sequencing analyses were performed in HepG2 cells transfected with control or β-catenin siRNA. A total of 76 candidate genes were selected based on the criterion that they were abundantly expressed (FPKM ≥ 3) and that levels decreased by more than threefold in β-catenin-depleted cells compared with control cells (Fig. 1a). Whether the candidate genes possess the DNA-binding sites of *TCF7L2* (TCF4) was determined by chromatin immunoprecipitation (ChIP)-sequencing in HepG2 cells using a gene set of ENCODE Transcription Factor Binding Site Profiles (TCF7L2_HepG2_hg19_1), and the criteria identified 11 genes (Fig. 1a and Supplementary Table 1). Most of the selected genes, including NKD1, LGR5, SP5, ZNRF3, RNF43, Axin2, CCND1, and DKK1, were well-known target genes of Wnt/β-catenin signaling. Therefore, GREB1 was further analyzed because whether GREB1 acts downstream of Wnt/β-catenin signaling has not yet been established.

The ChIP assay revealed that TCF4 and β-catenin form a complex with the putative TCF4-binding site in the 5′-upstream region −443 to −448 of the human *GREB1* gene (Fig. 1b). β-Catenin knockdown decreased GREB1 expression, as well as Axin2, at mRNA and protein levels in HepG2 cells (Fig. 1c). However, *GREB1* expression was unaffected by treatment with the ER degrader ICI.182.780 (Supplementary Fig. 1a). A GSK-3 inhibitor CHIR99021, which activates the β-catenin pathway and increases *Axin2* mRNA expression, induced GREB1 expression in a bipotential mouse embryonic liver (BMEL) cell line and β-catenin knockdown suppressed GREB1 expression in Huh6 cells (Supplementary Fig. 1b, c), another HB cell line that harbors G34V somatic activating mutation in *CTNNB1* gene[13] and expresses GREB1 less than HepG2 cells (Supplementary Fig. 1d). In ER-positive MCF7 breast cancer cells, GREB1 expression was dramatically decreased by treatment with ICI.182.780 and also reduced by in response to CHIR99021 (Supplementary Fig. 1e), indicating that HepG2 and MCF7 cells regulate GREB1 expression in a different manner. Furthermore, CHIR99021 treatment did not induce GREB1 expression in HLE, SNU387, SNU449, and Huh7, human HCC cell lines in which GREB1 was little expressed (Supplementary Fig. 1f). Taken together, these results suggest that GREB1 is a specific downstream target gene of Wnt/β-catenin signaling in HB and immature liver progenitor cells.

HB is a rare disease and 11 cases were analyzed immunohistochemically (Supplementary Table 2). GREB1 was detected in the nucleus of tumor lesions of 10 HB cases (90.9%), while it was not detected in the nontumor regions (Fig. 1d). β-catenin was detected in the cytoplasm and/or nucleus of the tumor lesions of nine cases (81.8%), and all nine cases were also positive for GREB1 (Supplementary Fig. 1g and Supplementary Table 2). The region-specific expression pattern of GREB1 in HB tissue tends to be positively correlated with the accumulation of β-catenin in consecutive sections (Fig. 1e). There were two types of tumors in HB tissues. One showed solid structures with unpolarized cells, and the other had tubular structures with polarized cells. GREB1 was specifically expressed in the former tumors (Supplementary Fig. 1h).

*GREB1* gene expression and its correlation with expression of target genes of Wnt/β-catenin signaling in HB were analyzed using a public dataset of HB patients (GEO ID: gse75271)[14]. Both parameters were available for 50 tumor lesions and five nontumor regions in HB cases. In the tumor lesions, *GREB1* mRNA was significantly upregulated compared with the nontumor regions (Fig. 1f). In addition, there was a significant positive correlation between expression levels of *GREB1* and those of target genes of Wnt/β-catenin signaling, such as *Axin2*, *DKK1*, *NKD1*, and *glutamine synthetase* (*GS*) in HB (Fig. 1g). However, expression of target genes of ER signaling, including *PRLR*, and *XBP1*, did not significantly correlate with *GREB1* expression (Supplementary Fig. 1i). These results support the idea that activation of

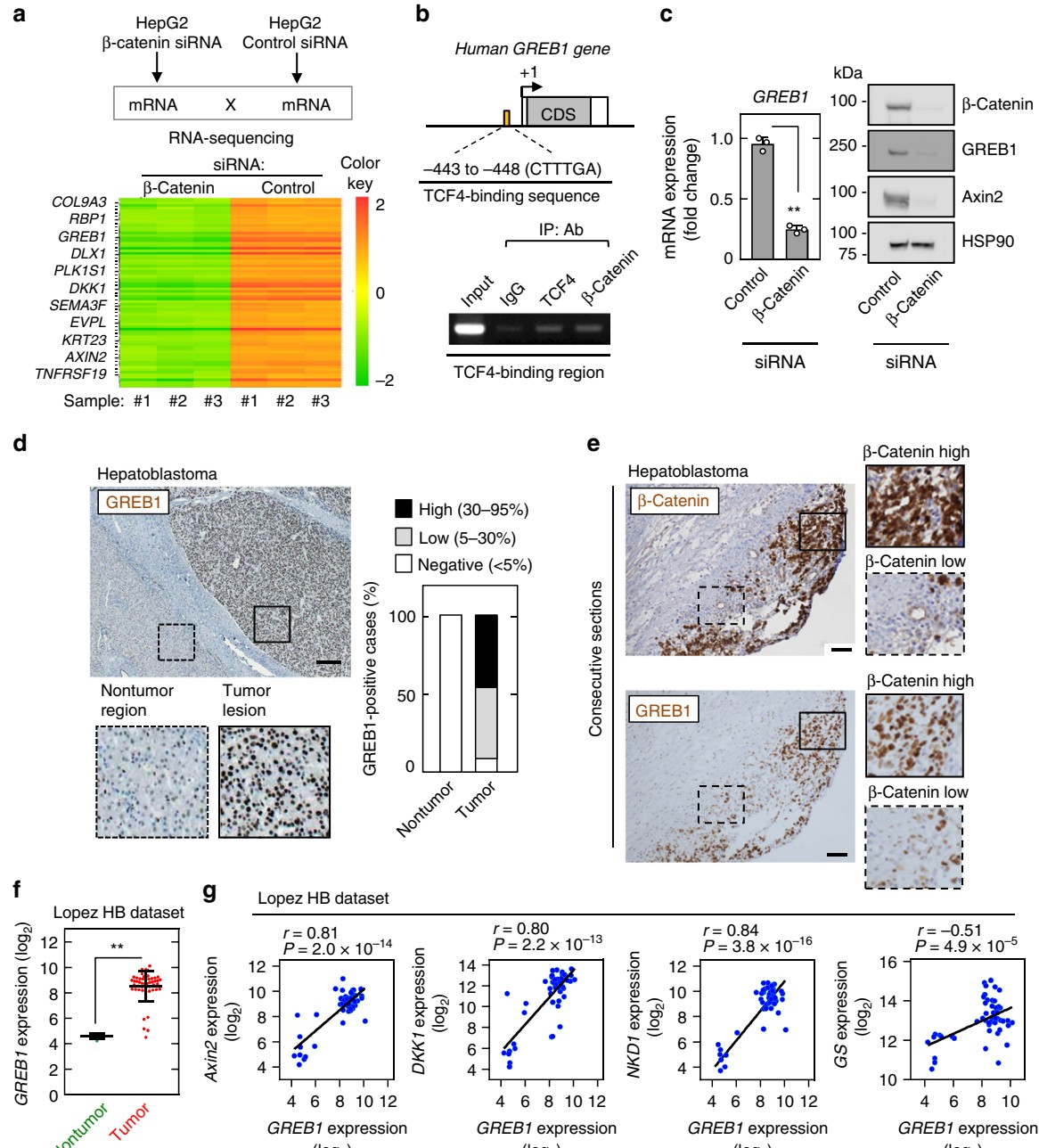

**Fig. 1** GREB1 is a downstream target gene of Wnt signaling in HB. **a** Downstream target genes of Wnt/β-catenin signaling in HB cells were identified by RNA-sequencing analysis. The heatmap of genes which has changed in expression in control and β-catenin knockdown cells and some candidate genes are shown in the lower panel. **b** Chromatin from HepG2 cells was immunoprecipitated with the indicated antibodies. The putative TCF4-binding region (−443 to −448) of the precipitated *GREB1* gene was analyzed by PCR with region-specific primers. **c** HepG2 cells were transfected with control or β-catenin siRNA and real-time PCR analysis for *GREB1* mRNA expression was performed. The obtained results are expressed as fold changes compared with control cells and the results shown are means ± SD. Whole lysates of control and β-catenin knockdown HepG2 cells were probed with the indicated antibodies. **d** HB tissues (*n* = 11) were stained with anti-GREB1 antibody and hematoxylin. Percentages of GREB1-positive cases in the tumor lesions and nontumor regions are shown in the right panels. Areas that stained positive for GREB1 were classified as indicated. Solid and dashed squares show enlarged images. **e** HB specimens were stained with the indicated antibodies and hematoxylin. Solid and dashed squares show the region where β-catenin staining is high and low, respectively. **f** *GREB1* gene expression in 5 nontumor and 50 HB cases, which were obtained from public mRNA profile dataset of HB, was analyzed. Results shown are scatter plots with means ± SD. **P < 0.01, t test. **g** Scatter plots showing correlation between target genes of the Wnt/β-catenin pathway (*Y*-axis) and *GREB1* gene expression (*X*-axis) were obtained from the dataset of HB. The solid line indicates linear fit. *r* indicates Pearson's correlation coefficient. *r* value and *P* value were calculated with GraphPad Prism 7. Scale bars in **d**, 200 μm; in **e**, 50 μm

Wnt/β-catenin signaling, but not that of ER signaling, induces *GREB1* expression in HB patients.

The public dataset of HB patients also revealed that *GREB1* expression is unchanged between HB cases with WT *CTNNB1* and those with *CTNNB1* mutations and deletions at exons 3 and 4 (Supplementary Fig. 1j)[14]. Expression of β-catenin target genes such as *GS* and *LGR5* expressions was also unchanged in HB cases regardless of the presence or absence of *CTNNB1*

abnormalities (Supplementary Fig. 1j). Therefore, *GREB1* expression in HB is correlated with β-catenin signaling activity but not always be associated with mutations and deletions at exons 3 and 4 of *CTNNB1*. Other *CTNNB1* mutations or *CTNNB1* mutation-independent activation of β-catenin signaling could occur in these cases.

**GREB1 expression is involved in HB cell proliferation.** β-Catenin knockdown decreased HepG2 cell proliferation as reported[15], and exogenous GREB1 expression partially rescued the phenotype (Supplementary Fig. 2a). To examine the roles of GREB1 in HB cell proliferation, GREB1 was depleted by two different siRNAs, which target the 3′-UTR (Supplementary Fig. 2b). GREB1 knockdown did not affect expression of β-catenin and Axin2 at protein levels or that of *PRLR* and *XBP1* at mRNA levels (Supplementary Fig. 2b, c), suggesting that GREB1 does not function upstream of Wnt/β-catenin and ER in HB cells. GREB1 knockdown decreased HepG2 cell proliferation using two-dimensional (2D) culture conditions, and exogenous GREB1 expression rescued the GREB1-knockdown phenotypes (Fig. 2a).

When HepG2 cells were grown using 3D culture conditions with Matrigel, GREB1 knockdown reduced the area of tumor sphere (Fig. 2b). Conversely, micro-lumens were frequently observed in GREB1-depleted spheres, suggesting that HepG2 cells are transformed into a differentiated state with epithelial polarization (Fig. 2b). These results are consistent with the observation that GREB1 was specifically expressed in the solid structure tumors with unpolarized cells in HB tissues (see Supplementary Fig. 1h). GREB1 knockdown also decreased expression of a hepatoblast marker *AFP*, an imprinted gene *PEG3*, and a HB marker *DLK1* in HepG2 cells (Fig. 2c). Consistently, there were strong positive correlations between expression levels of *GREB1* and those of *DLK1* and *TACSTD1* (another HB marker gene) in HB (Supplementary Fig. 2d). Furthermore, GREB1 overexpression in HepG2 and Huh6 cells promoted cell proliferation (Fig. 2a and Supplementary Fig. 2e,f).

GREB1 knockdown also decreased protein expression of cell cycle markers, including cyclinA, cyclinB, and phosphorylated histoneH3 (Fig. 2d). The public dataset revealed a positive correlation between mRNA expression levels of *GREB1* and *MKI67*, as well as *GMMN* and *PCNA*, which are involved in cell cycle progression (Fig. 2e). These knockdown phenotypes including cell proliferation, HB marker genes expression, and cell cycle progression were confirmed using GREB1-knockout HepG2 cells, which were generated using a CRISPR/Cas9 system, and GREB1 expression rescued them (Supplementary Fig. 2g–j).

Furthermore, GREB1 knockdown showed an increase in cell death indicated by propidium iodide staining in HepG2 cells, and the caspase inhibitor Z-VAD suppressed GREB1 knockdown-induced cell death (Fig. 2f). Consistently, intracellular levels of cleaved caspase 3 and PARP1 increased in GREB1-depleted HepG2 cells (Fig. 2g). Taken together, these results demonstrate that GREB1 is not only necessary for cellular proliferation but also for cell survival of HB cells.

**GREB1 forms a complex with Smad2/3.** HA-FLAG-GREB1 was mainly localized to the nucleus of X293T cells (Fig. 3a). When the nuclear localization sequence (NLS) (amino acids 310–319) was deleted, HA-FLAG-GREB1-ΔNLS was present throughout the cytosol (Fig. 3a). To clarify how GREB1 regulates cellular proliferation of HB cells, possible interacting proteins were retrieved using the BioGRID database (https://thebiogrid.org/) (Supplementary Table 3). Among the possible GREB1-binding proteins, we focused on Smad4, a central mediator for TGFβ signaling[16]. HA-FLAG-GREB1 formed a complex with GFP-tagged Smad3

(Co-Smad) and Smad7 (I-Smad), but not Smad4 (R-Smad) in X293T cells (Fig. 3b). Conversely, HA-FLAG-GREB1-ΔNLS associated with GFP-tagged Smad4 but not Smad3 or Smad7 (Fig. 3b). Since GFP-Smad3 and GFP-Smad7 were localized to the nucleus while GFP-Smad4 was localized to the cytosol in X293T cells (Supplementary Fig. 3a), these results indicate that GREB1 bound to all these Smad family members. In HepG2 cells, GREB1 bound to Smad2/3 endogenously (Fig. 3c), but the interaction of GREB1 with Smad7 or Smad4 was hard to detect. GFP-GREB1 expressed in Huh6 cells also formed a complex with endogenous Smad2/3 (Supplementary Fig. 3b). Other nuclear proteins, β-catenin and c-Myc, did not interact with GFP-Smad3, 4, or 7 (Supplementary Fig. 3c).

Smads have two functional regions, MH1 and MH2 (Fig. 3d)[17]. The C-terminal region of GFP-Smad2 containing the MH2 domain (GFP-Smad2/C) (266–467) formed a complex with HA-FLAG-GREB1 in X293T cells, but GFP-Smad2/N (1–265) did not (Fig. 3d). The MH2 domain, but not the MH1 domain, of recombinant GST-Smad2 bound to endogenous GREB1 in HepG2 and Huh6 cell lysates (Fig. 3e and Supplementary Fig. 3d) under the conditions that GST-Smad2 MH2 domain formed a complex with Smad4. Taken together, these results suggest that GREB1 directly interacts with Smad2/3 in the nucleus of HB cells.

To analyze which region of GREB1 interacted with Smad2/3, GREB1 deletion mutants divided into three regions (N, M, and C) were generated. Three copies of the NLS were fused to the N-terminus of the GFP-GREB1/M (667–1333) and GFP-GREB1/C (1334–1954) (Fig. 3f). GFP-GREB1/N (1–666) and the NLS-fused deletion mutants of GREB1 were found to be localized to the nucleus (Fig. 3f). Although all the deletion mutants showed less-binding activity to FLAG-Smad3 compared with GFP-GREB1 (WT), NLS-GREB1/M had relatively high-binding affinity (Fig. 3g). Not only NLS-GREB1 lacking the middle region (667–1333) (GFP-GREB1/ΔM) but also deletion of N-terminal or C-terminal region of GREB1, decreased the binding activity of GREB1, suggesting that an intact structure of GREB1 is required for full-binding activity to the Smad2/3 (Supplementary Fig. 3e).

**GREB1 acts as a negative regulator of TGFβ signaling.** Dataset of HB patients showed that *Axin2* and *DKK1* mRNAs were upregulated, while *PAI-1* and *GADD45B* (target genes of TGFβ signaling) mRNAs were significantly downregulated in HB lesions compared with the nontumor regions (Fig. 4a; Supplementary Fig. 4a). The dataset also revealed that there was a significant inverse correlation between expression levels of *GREB1* mRNA and those of target genes of TGFβ signaling, such as *PAI-1 GADD45B*, *p21/CDKN1A*, and *TSP1* (Fig. 4b). Consistently, GREB1 knockdown in HepG2 cells increased expression of TGFβ signal target genes, such as *PAI-1* and *SNAIL2*, and the phenotypes were rescued by GFP-GREB1 expression (Fig. 4c). *PAI-1* mRNA was also increased in GREB1 KO HepG2 cells and GREB1 expression restored it (Supplementary Fig. 4b). In addition, ALK5 inhibitor counteracted upregulation of *PAI-1* and *SNAIL2* mRNAs induced by GREB1 knockdown (Fig. 4d). However, GREB1 knockdown did not affect TGFβ-dependent nuclear translocation, binding of Smad2/3 to Smad4, or phosphorylation of Smad2/3 (Supplementary Fig. 4c–e). Therefore, GREB1 may act downstream of nuclear translocation of Smad2/3 in TGFβ signaling.

Transcriptional coactivators such as CBP and p300 possess histone acetyltransferase activity to modify chromatin structure[18]. The MH2 domain of R-Smads (Smad2/3) directly interacts with CBP or p300 and the interaction is required for the transcriptional activation function of R-Smads[19]. Endogenous interaction of Smad2/3 and p300 was increased by GREB1 knockdown in

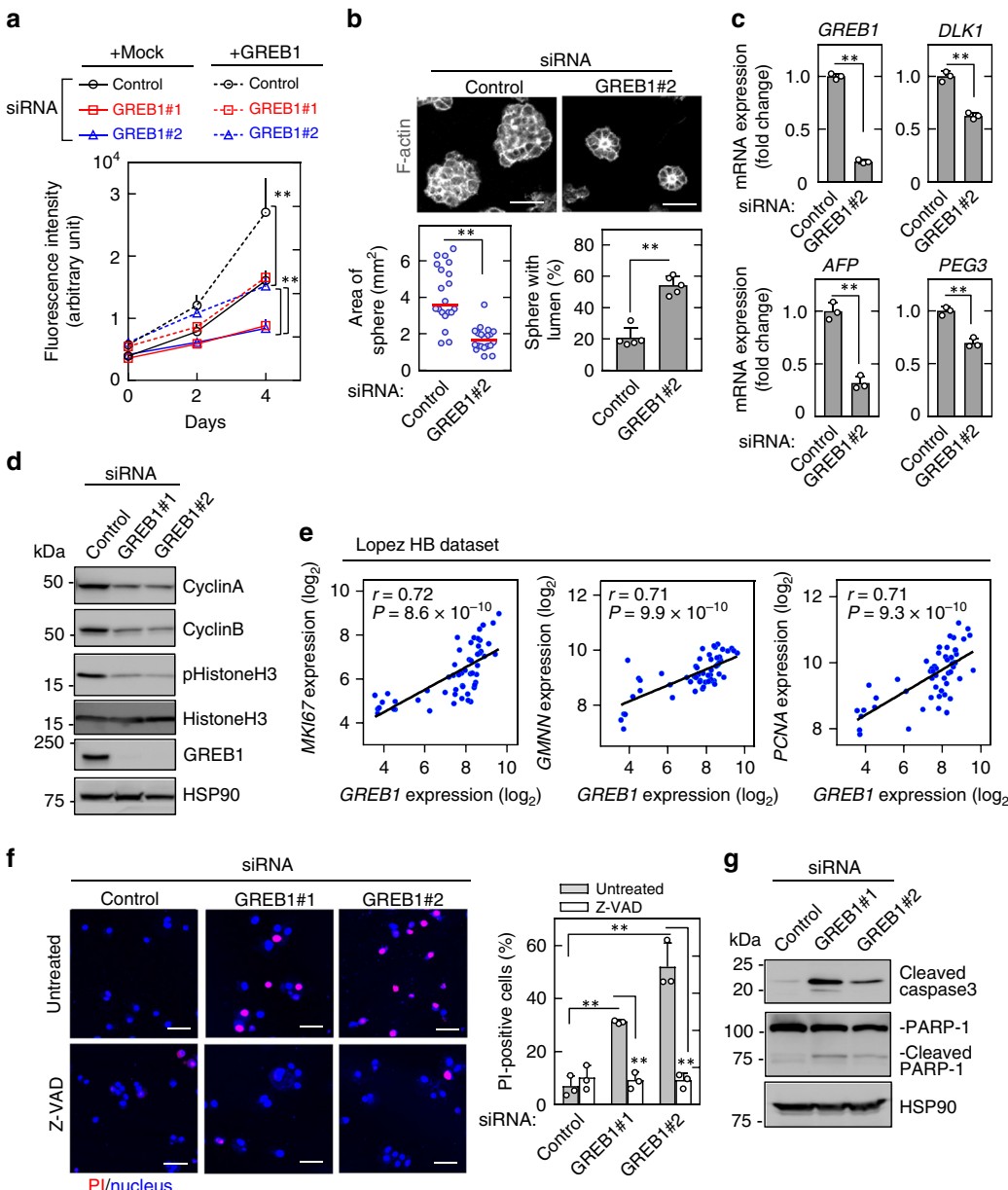

**Fig. 2** GREB1 expression is involved in HB cell proliferation in vitro. **a** HepG2 cells expressing mock or GREB1 were transfected with control or two independent GREB1 siRNAs and cultured on a 2D plastic dish for the indicated numbers of days. Relative cell numbers were quantified using the Cyquant assay. Results are shown as means ± SD. **P < 0.01, ANOVA and post hoc test. **b** HepG2 cells transfected with control or GREB1 #2 siRNA were cultured for 5 days in 3D Matrigel. Cells were then stained with phalloidin, and areas of spheres were calculated; results are shown as dot plots with median (n = 20). Spheres with polarized lumen are expressed as the percentage of spheres with F-actin-positive central microlumen compared with the total spheres. The results shown are means ± SD (n = 50). **P < 0.01, t test. **c** HepG2 cells were transfected with control or GREB1 #2 siRNA and real-time PCR experiments for indicated mRNA expression were performed. **P < 0.01, t test. **d** HepG2 cells were transfected with control or two independent GREB1 siRNAs and cultured in medium with 0.1% FBS for 24 h. Lysates were probed with the indicated antibodies. **e** Scatter plots showing correlation between marker genes of cell proliferation (Y-axis) and GREB1 gene expression (X-axis) were obtained from the dataset of HB. r value and P value were calculated with GraphPad Prism 7. **f** HepG2 cells were transfected with control or two independent GREB1 siRNAs and cultured in medium with 0.1% FBS for 48 h with or without Z-VAD. Cells were incubated with PI and Hoechst33342. PI-positive cells are expressed as the percentage of positively stained cells compared with total Hoechst33342 stained cells per field (n = 100–200). **P < 0.01, ANOVA and post hoc test. **g** HepG2 cells were transfected with control or two independent GREB1 siRNAs and cultured in medium with 0.1% FBS for 48 h, and the lysates were probed with the indicated antibodies. Scale bars in **b**, 50 μm; in **f**, 100 μm

HepG2 cells (Fig. 4e). GFP-Smad2/C containing the MH2 domain interacted with p300 in X293T cells, and overexpression of HA-FLAG-GREB1 inhibited their interaction (Fig. 4f), suggesting that GREB1 competes with p300 for binding to Smad2/3. The ChIP assay revealed that GREB1 knockdown increased levels of acetylated histone H4 (acH4) at the PAI-1 locus (exon2) of

HepG2 cells, irrespective of exogenous TGFβ stimulation (Fig. 4g). Expression of constitutively active mutant of TGFβ receptor 1 (TGFBR1[T204D]) increased the SNAIL2 and p15/CDKN2B (target gene of TGFβ signaling) mRNA levels, and GFP-GREB1 but not GFP-GREB1/ΔM lacking the Smad2/3-binding region inhibited the upregulation (Fig. 4h). Under the same conditions, GREB1

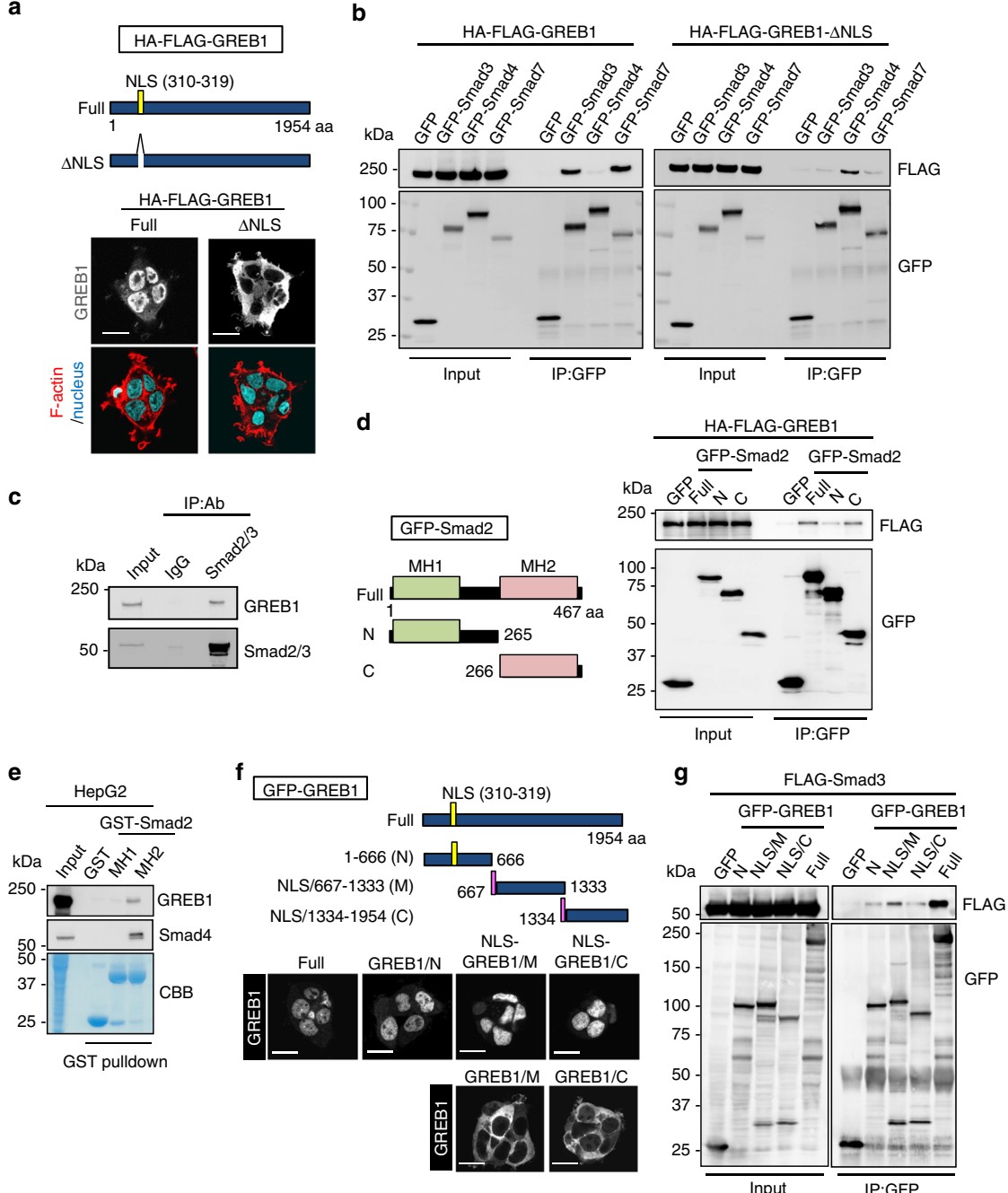

**Fig. 3** GREB1 forms a complex with Smad2/3. **a** Schematic representation of the GREB1 and GREB1-ΔNLS (310–319) are shown. X293T cells expressing the indicated proteins were fixed and stained with anti-FLAG antibody, phalloidin, and Hoechst33342. **b** Lysates of X293T cells expressing the indicated proteins were immunoprecipitated with anti-GFP antibody, and the immunoprecipitates were probed with the indicated antibodies. **c** Lysates of HepG2 cells were immunoprecipitated with anti-Smad2/3 antibody and the immunoprecipitates were probed with the indicated antibodies. **d** Schematic representation of the deletion mutants of Smad2 used in this study is shown. Lysates of X293T cells expressing HA-FLAG-GREB1 and GFP-Smad2 mutants were immunoprecipitated with anti-GFP antibody and the immunoprecipitates were probed with the indicated antibodies. **e** Lysates of HepG2 cells were precipitated with recombinant GST, GST-Smad2/MH1, or GST-Smad2/MH2 proteins and the precipitates were probed with the indicated antibodies. **f** Schematic representation of three deletion mutants (N, M, and C) of GFP-GREB1 is shown. HepG2 cells expressing deletion mutants of GFP-GREB1 were fixed and stained with anti-GFP antibody. **g** Lysates of X293T cells expressing FLAG-Smad3 and GFP-GREB1 mutants were immunoprecipitated with anti-GFP antibody and the immunoprecipitates were probed with the indicated antibodies. Scale bars in **a** and **f**, 20 μm

overexpression did not affect *Axin2* mRNA expression (Fig. 4h). Consistent with the observation that TGFβ signaling suppresses AFP and DLK1 in rat fetal hepatocytes and HCC cells[20,21], TGFBR[T204D] expression in HepG2 cells indeed decreased the levels of *AFP* and *DLK1* mRNAs; Smad2/3 knockout rescued GREB1 knockdown-induced decrease in *AFP* expression

(Supplementary Fig. 4f–h). Therefore, GREB1 may prevent Smad2/3 from binding to p300 and inhibit expression of target genes of TGFβ-Smad signaling. However, since Smad2/3 knockout did not rescue *DLK1* gene suppression induced by GREB1 knockdown, the mechanism of TGF-β signal-mediated regulation of DLK1 expression remains elusive.

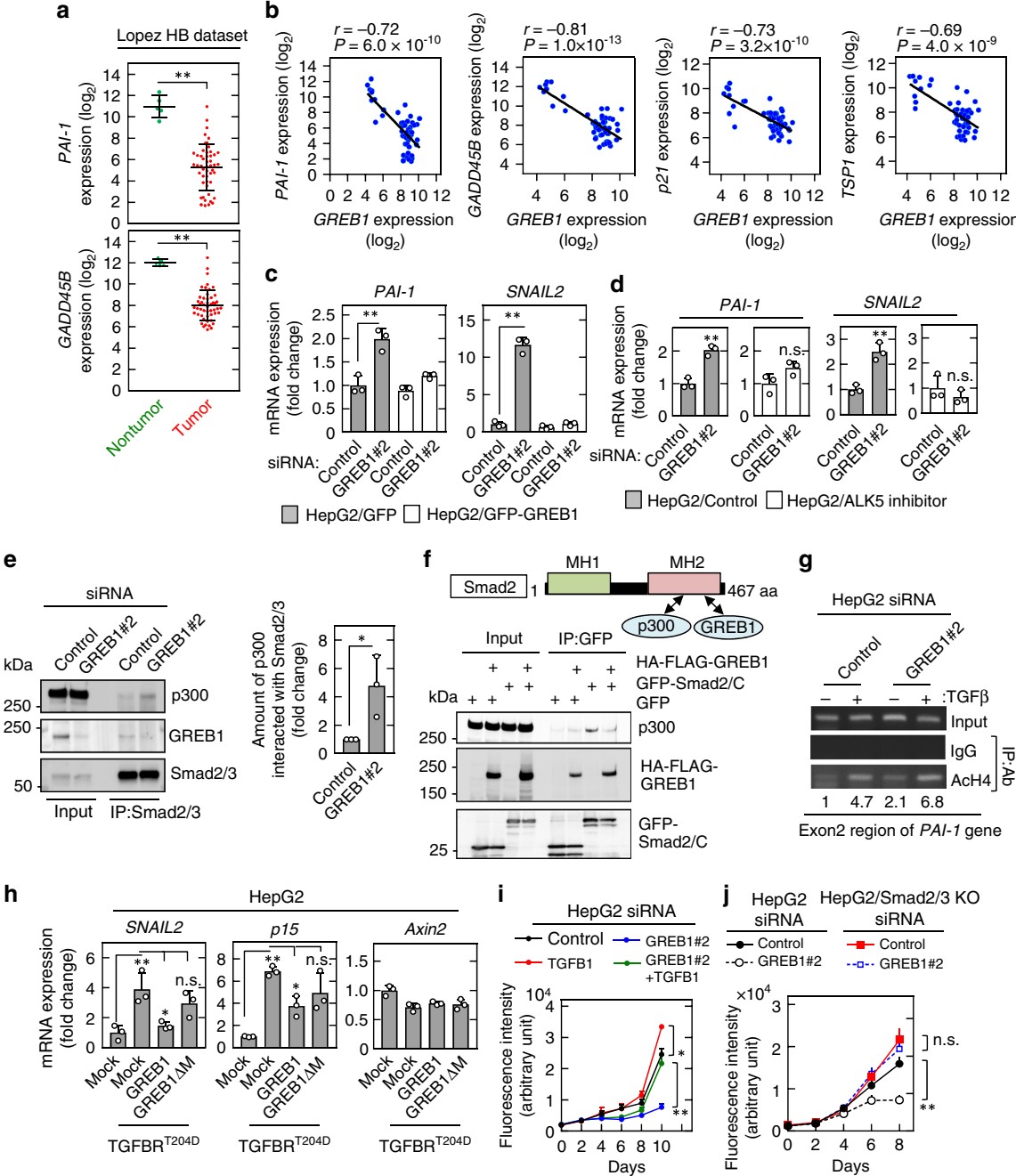

**Fig. 4** GREB1 inhibits TGFβ signaling. **a** Expression of *PAI-1* and *GADD45B* gene, which were obtained from dataset of HB, was analyzed. The results shown are scatter plots with means ± SD. **P < 0.01, t test. **b** Scatter plots showing correlation between target genes of TGFβ signaling (Y-axis) and *GREB1* gene expression (X-axis) were obtained from the dataset of HB. r value and P value were calculated with GraphPad Prism 7. **c** HepG2 cells stably expressing GFP or GFP-GREB1 was transfected with control or GREB1 #2 siRNA. Real-time PCR analyses for indicated mRNA expression were performed. **d** HepG2 cells were transfected with control or *GREB1* #2 siRNA with or without ALK5 inhibitor and real-time PCR experiments for indicated mRNA expression were performed. **P < 0.01, t test. **e** Lysates of HepG2 cells transfected with control or *GREB1* #2 siRNA were immunoprecipitated with anti-Smad2/3 antibody. The immunoprecipitates were probed with the indicated antibodies. Relative band intensities of p300 coprecipitated with Smad2/3 were quantified. **P < 0.01, t test. **f** The binding region of Smad2 for p300 and GREB1 is shown. Lysates of HepG2 cells expressing the indicated proteins were immunoprecipitated with anti-GFP antibody. The immunoprecipitates were probed with the indicated antibodies. **g** Chromatin from HepG2 cells transfected with control or *GREB1* #2 siRNA and untreated or treated with 10 ng/ml TGFβ1 for 30 min was immunoprecipitated with IgG or anti-acH4 antibody. The exon2 region of the precipitated *PAI-1* gene was analyzed by PCR with region-specific primers. **h** HepG2 cells were transfected with the indicated plasmids. Real-time PCR experiments for indicated mRNA expression were performed. **P < 0.01; *P < 0.05, ANOVA and post hoc test. **i** HepG2 cells were transfected with the indicated siRNAs, and cultured for the indicated numbers of days. Relative cell numbers were quantified using the Cyquant assay. **P < 0.01, ANOVA and post hoc test. **j** Control or Smad2/3 KO HepG2 cells were transfected with control or GREB1 #2 siRNA and cultured for the indicated numbers of days. Relative cell numbers were quantified using the Cyquant assay. n.s., not significant. **P < 0.01; *P < 0.05, ANOVA and post hoc test

Based on RNA-seq data from the cancer cell line encyclopedia, HepG2 cells expressed much higher levels of *TGFB1* than *TGFB2* or *TGFB3* (Supplementary Fig. 5a). Knockdown of both TGFβ1 and GREB1 canceled upregulation of *PAI-1* mRNA by GREB1 knockdown (Supplementary Fig. 5b). Knockdown of TGFβ1 and GREB1 increased and decreased, respectively, HepG2 cell proliferation, and knockdown of TGFβ1 in GREB1-depleted cells rescued the growth suppression (Fig. 4i and Supplementary Fig. 5c). GREB1-depleted cells exhibiting normal cell growth and *PAI-1* expression by additional knockdown of TGFB1, showed an increased response of *PAI-1* expression to TGFβ1 stimulation in the range from 0.01 to 1 ng/ml compared with cells transfected with control or TGFB1 siRNA alone (Supplementary Fig. 5d). Among TGFβ-regulated cytostatic genes, including *p21*, *p15*, and *p27*, *p15* was dramatically increased by GREB1 knockdown in HepG2 cells and the phenotype was rescued by Smad2/3 knockout (Supplementary Fig. 5e, f). p15 knockdown rescued GREB1 depletion-induced growth suppression and cell death (Supplementary Fig. 5g, h), suggesting that p15 is a key mediator of both growth inhibition and cell cycle arrest, followed by cell death. Consistently, Smad2/3 knockout inhibited the decrease in cell proliferation induced by GREB1 knockdown (Fig. 4j). Thus, GREB1 depletion may increase the susceptibility to autocrine TGFβ signaling, resulting in the inhibition of cell proliferation.

Treatment with ICI.182.780 or GREB1 knockdown increased *PAI-1* mRNA expression in MCF7 breast cancer cells (Supplementary Fig. 5i, j) and GREB1 formed a complex with Smad2/3 in MCF7 cells (Supplementary Fig. 5k), suggesting that GREB1 functions as a negative regulator of TGFβ signaling not only in HB but also breast cancer cells.

**Binding of GREB1 to Smad2/3 inhibits transcription**. GFP-GREB1 was observed as punctates in the interchromatin compartment rather than the chromatin territory of the nucleus of HepG2 cells (Supplementary Fig. 6a). The GREB1 punctates were not colocalized with fibrillarin, SC35, PML protein, or coilin (Supplementary Fig. 6b), suggesting that GREB1 foci are independently present of nucleoli, nuclear speckles, PML bodies, and Cajal bodies. GFP-GREB1 and FLAG-Smad3 were colocalized and the complex was observed in the interchromatin compartment (Supplementary Fig. 6c). Endogenous GREB1 and Smad2/3 were present in both the cytoplasm and nuclei of HB cells, and TGFβ stimulation led to dominant accumulation in the nucleus (Fig. 5a). Phosphorylated Smad2/3 clearly formed foci in the nucleus of TGFβ-treated cells and colocalized with GREB1 (Fig. 5b). In addition, PLA assays revealed that TGFβ stimulation led to close association of Smad2/3 with GREB1 in the border zone between the chromatin territory and interchromatin compartment (Fig. 5c). GFP-GREB1/N or GFP-NLS-GREB1/M was present throughout the nucleus but did not form foci, whereas GFP-NLS-GREB1/C was observed as foci as well as GFP-GREB1 (Fig. 5d), suggesting that the C-terminal region is critical for the localization of GREB1 to the spatially specific region of the nucleus.

Ethynyl uridine (EU) labeling to assess RNA synthesis was detected in the interchromatin region in HepG2 cells (Supplementary Fig. 6d). Nascent RNA molecules were observed in the interchromatin compartment of HepG2 cells expressing GFP-Smad3, some of which were colocalized with nuclear foci of GFP-Smad3 (Fig. 5e). Conversely, when HA-FLAG-GREB1 was expressed, EU labeling was decreased on the nuclear foci of GFP-Smad3 (Fig. 5e). These results suggest that transcriptional activity associated with TGFβ-Smad signaling is selectively suppressed by the interaction of Smad2/3 with GREB1 in the border zone of the chromatin and interchromatin regions.

**GREB1 is involved in HB-like tumor formation in vivo**. Several types of mouse liver tumor models for HCC and HB were developed[22–24]. Overexpression of constitutively active forms of β-catenin and YAP using hydrodynamic transfection generates liver tumors with characteristics of HB and HCC[25]. YAP was indeed detected in the tumor lesions of 9 (81.8%) out of our 11 HB cases (Fig. 6a), and all nine cases were also positive for GREB1 and β-catenin (Supplementary Table 2). In addition, the HGF-c-Met pathway is activated in HB[26].

Which combination of β-catenin, YAP, and c-Met is effective in generating the HB model was tested. In our conditions of hydrodynamics transfection, YAPS127A and ΔNβ-catenin (BY model) induced small solid liver nodules but the nodules little expressed GREB1 and DLK1 (Fig. 6b). The combination of c-Met and ΔNβ-catenin (BM model) induced liver nodules expressing GREB1 and DLK1 to the small extent (Fig. 6b)[22]. In contrast, the combination of ΔNβ-catenin, YAPS127A, and c-Met (BYM model) induced larger and multiple liver nodules throughout the liver surface and the tumor nodules highly expressed both GREB1 and DLK1 (Fig. 6b). Tumor nodules of BYM model expressed mRNAs of *GREB1* and *TACSTD1* higher than those of BY or BM model (Fig. 6c). Therefore, the BYM model seemed to be appropriate for in vivo analyses of GREB1 functions in HB.

Nuclear YAP accumulation was observed in Huh6 cells more significantly than HepG2 cells (Supplementary Fig. 7a). However, knockdown of YAP and TAZ in HepG2 and Huh6 cells did not affect GREB1 expression (Supplementary Fig. 7b). In addition, treatment of Huh6 cells with CHIR99021 and/or XMU-MP-1, which inhibits Mst1/2 kinase to activate YAP[27], increased expression of *Cyr61*, a YAP target gene, but rather decreased *GREB1* expression as well as *Axin2* (Supplementary Fig. 7c). Therefore, YAP and TAZ are necessary for promoting HB tumorigenesis but is not essential for GREB1 expression.

c-Met knockdown in HepG2 cells decreased expression of *Axin2* and *GREB1* mRNAs, but rather increased expression of *ANKRD1* and *Cyr61* mRNAs (Supplementary Fig. 7d). It is reported that the HGF-c-Met pathway phosphorylates β-catenin at tyrosine 654 and induces β-catenin nuclear translocation and activation[28,29]. c-Met knockdown indeed suppressed the phosphorylation level of β-catenin at Tyr654 and expression of GREB1 and Axin2 (Supplementary Fig. 7e). Thus, c-Met would activate β-catenin signaling, thereby increasing GREB1 expression.

HB-related gene expression and histological appearance of six tumor nodules taken from each BYM mouse (C1-C7) were investigated. The volume of whole livers and serum AFP values in the mice of BYM model dramatically increased compared with WT mice (Fig. 7a, b). Real-time PCR analyses revealed that BYM-tumor nodules express all three exogenous genes although the expression levels were variable (Supplementary Fig. 8a). The *GREB1* mRNA level in individual tumor nodule was higher than that in the nontumor regions and normal liver tissue (Supplementary Fig. 8b). Heatmap visualization revealed that *GREB1* mRNA levels vary according to tumor nodules in individual mice, and we classified the tumors into high (C1, C4, and C6) and low (C2, C3, C5, and C7) expression groups. In the high GREB1 expression group, *TACSD1*, *DLK1* (HB marker genes), *AFP*, *GPC3* (undifferentiated hepatoblast marker genes), *PEG3*, *MEG3*, *BEX1* (imprinted genes), and *Axin2* tended to be higher than the low GREB1 expression group (Fig. 7c). There was a strong positive correlation between the expression levels of *GREB1* and HB-related genes such as *DLK1*, *TACSTD1*, *GPC3*, *MEG3*, and *Axin2* (Supplementary Fig. 8c and Supplementary Table 4).

Most tumors from the high GREB1 group of mice showed solid growth patterns with a high nuclear: cytoplasmic ratio and some tumors included a macrotrabecular pattern (Fig. 7 and Supplementary Fig. 8d). Tumors from the low GREB1 group of

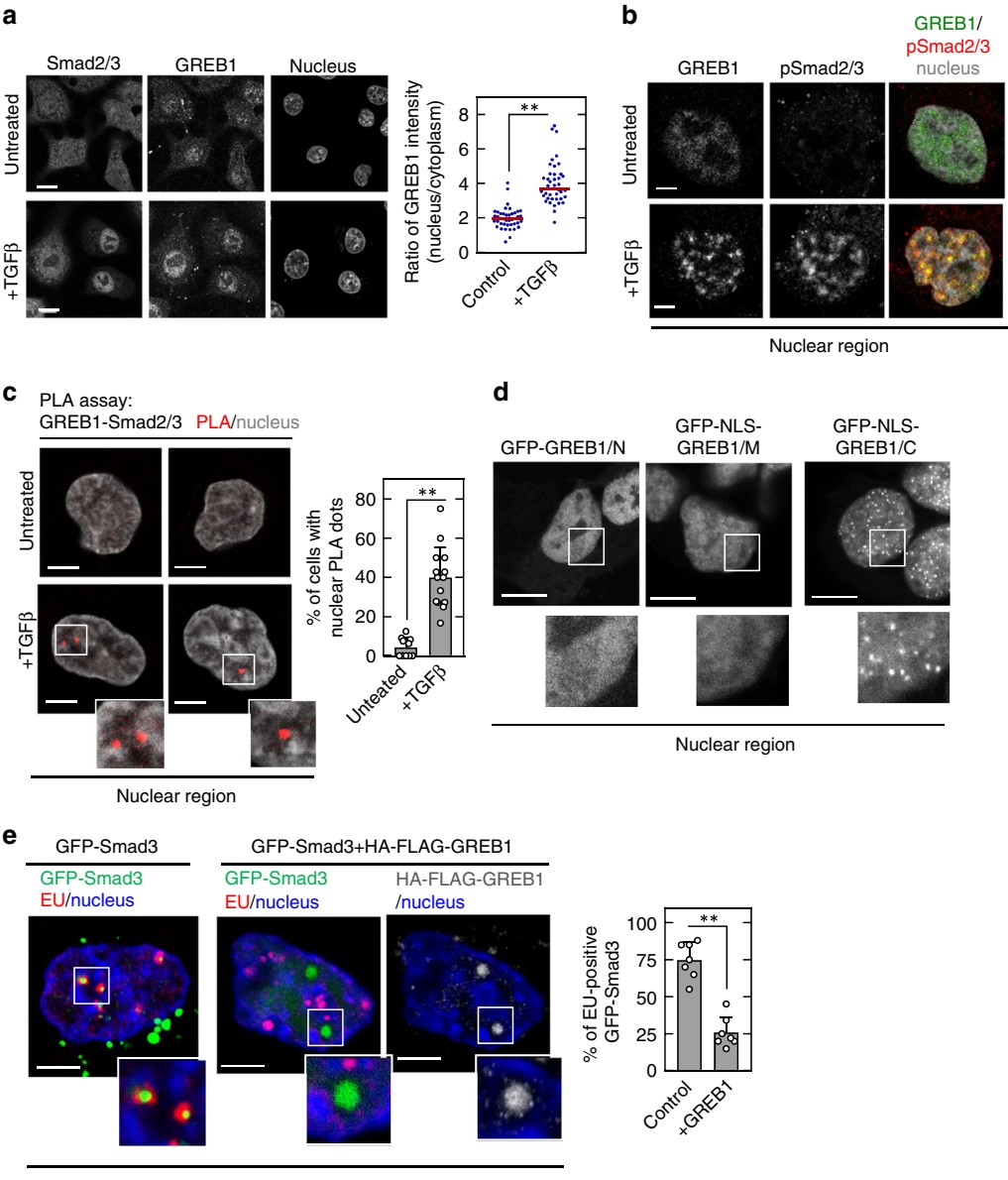

**Fig. 5** Interaction of GREB1 and Smad2/3 in the nucleus inhibits Smad-dependent transcription. **a** HepG2 cells were untreated or treated with 10 ng/ml TGFβ for 30 min, followed by staining with the indicated antibodies and Hoechst33342. GREB1 fluorescence intensity in the nucleus and cytoplasm was measured and the results are expressed as the ratio of nuclear to cytoplasmic GREB1. **P < 0.01, t test. **b** HepG2 cells were untreated or treated with TGFβ, and cells were then stained with the indicated antibodies and Hoechst33342. **c** HepG2 cells were untreated or treated with TGFβ, followed by incubation with mouse anti-GREB1 and rabbit anti-Smad2/3 antibodies, which were then combined with secondary PLA probes. Interaction events are shown as red dots. Cells with PLA dots were counted, and the results are expressed as the percentage of total cells per field (n > 100). The regions in the solid squares are shown enlarged. **P < 0.01, t test. **d** HepG2 cells expressing deletion mutants of GFP-GREB1 were stained with anti-GFP antibody. The regions in the solid squares are shown enlarged. **e** HepG2 cells expressing GFP-Smad3 with or without HA-FLAG-GREB1 were incubated with EU for 30 min before fixation. Cells were stained with the indicated antibodies and Hoechst33342. The regions in the solid squares are shown enlarged. **P < 0.01, t test. Scale bars in **a** and **d**, 10 μm; in **b**, **c**, and **e**, 5 μm

mice mainly contained well-differentiated bigger cells with a clear cytoplasm, uniformly round nuclei, and small nucleoli (Fig. 7d; Supplementary Fig. 8d). Thus, targeted coexpression of ΔNβ-catenin, YAPS127A, and c-Met in mouse hepatocytes preferentially drives development of well to moderately-differentiated HB-like tumors, which are reminiscent of human crowded fetal HB[24]. Expression of GREB1 and DLK1 in the tumor lesions was higher than the nontumor regions, and their staining levels were correlated with the degree of differentiation (Fig. 7e).

Knockdown of GREB1 by shRNA in BYM mice (BYM GREB1 KD mice) (K1-K6), resulted in reduced tumor formation, and no tumors were present in four out of the six mice (Fig. 7a). Compared with control BYM mice, the liver weights and serum AFP values were dramatically decreased in BYM GREB1 KD mice (Fig. 7b). Tumors developed in BYM GREB1 KD mice (K2 and K4) that expressed low levels of *GREB1* mRNA, and their *DLK1* and *TACSTD1* expression levels were downregulated compared with the high GREB1 expression group of control BYM mice (Fig. 7f). These tumor cells contained well-differentiated HB-like

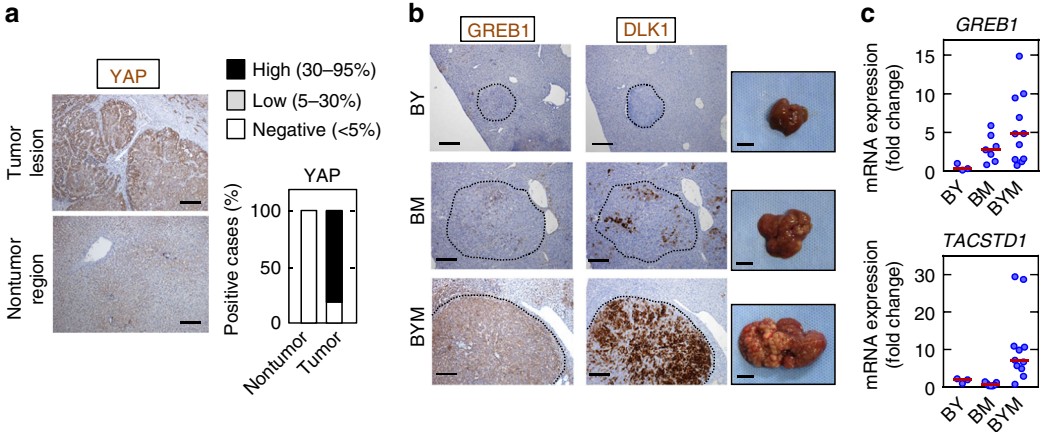

**Fig. 6** Forced expression of β-catenin, YAP, and c-Met induced HB-like tumorigenesis in vivo. **a** HB tissues (n = 11) were stained with anti-YAP antibody and hematoxylin. Percentages of YAP-positive cases in the tumor lesions and nontumor regions are shown. **b** Tissue sections of the liver isolated from mice injected with ΔNβ-catenin and YAPS127A (BY), ΔNβ-catenin and c-Met (BM), or ΔNβ-catenin, YAPS127A, and c-Met (BYM) at 6 weeks after hydrodynamic tail vein injection of plasmids, were stained with anti-GREB1, anti-DLK1, and hematoxylin. Solid squares showed the representative gross images of livers from mice injected with BY, BM, or BYM. **c** Total RNAs were prepared from liver tumor nodules isolated from BY (n = 3), BM (n = 7), or BYM (n = 11) mice. Real-time PCR experiments for GREB1 and TACSTD1 mRNA expression are performed. Results are expressed as fold changes compared with BY and are shown as dots plot with median. Scale bars in **a** and **b** (left six panels), 200 μm; in **b** (right three panels), 1 cm

cells with clear cytoplasm (Fig. 7g). The expression of N-cadherin, a target gene of TGFβ signaling, was lower in the tumor lesions compared with the nontumor regions of BYM mouse livers, whereas in BYM GREB1 KD mice livers, N-cadherin was upregulated in the tumor lesions and became comparable to the nontumor regions, suggesting that TGFβ signaling was activated by GREB1 depletion (Fig. 7h). Therefore, GREB1 is involved in HB-like histological patterns, marker gene expression, and tumor formation in this mouse model.

**GREB1 represents a molecular target for HB therapy**. While control HepG2 cells formed subcutaneous xenograft tumors, GREB1-knockout HepG2 cells resulted in smaller and lighter tumors (Fig. 8a). Expression of GREB1 rescued phenotypes induced by GREB1 knockout, but GREB1/ΔNLS did not (Fig. 8a). Furthermore, Smad2/3 knockout rescued impaired cell proliferation in vitro and xenograft tumor formation by GREB1 knockout (Fig. 8a; Supplementary Fig. 8e, f), suggesting that GREB1 abrogates TGFβ signal-dependent inhibition of cell proliferation.

AmNA-modified ASO shows a high nuclease resistance along with a high RNA selectivity, and has been shown to accumulate in the liver by systemic administration[30]. To examine the effects of AmNA-modified human GREB1 ASO on tumor growth, 20 GREB1 ASOs were synthesized (Supplementary Table 5); their ability to reduce GREB1 protein expression in HepG2 cells was evaluated. Of the 20 GREB1 modified ASOs, ASO-6434, ASO-6921 ASO-6968, and ASO-7724 strongly suppressed GREB1 expression in HepG2 cells without cytotoxicity (Supplementary Fig. 9a). These GREB1 ASOs suppressed the sphere formation activity of HepG2 cells. The phenotypes were rescued by GFP-GREB1 expression in cells treated with GREB1 ASOs-6921, −6968, and −7724, but not by ASO-6434 (Fig. 8b).

HepG2 cells were injected into the liver in a high concentration of Matrigel (day 0), and tumor formation was observed at day 27 after implantation. Starting from day 3, mice were subcutaneously injected twice a week with control ASO or GREB1 ASOs-6921 and −7724. Both GREB1 ASOs suppressed HepG2 tumor formation and reduced tumor weight compared with control ASO (Fig. 8c). GREB1 ASOs-6921 and −7724 also decreased the number of Ki-67-positive cells and increased the number of

apoptotic tumor cells (Fig. 8d, e). GREB1 expression was inhibited and PAI-1 gene expression tended to be increased by GREB1 ASOs-6921 and −7724 in HB liver tumors (Fig. 8f), suggesting activation of TGFβ signaling. These GREB1 ASOs did not induce histological damage or cell death in the nontumor regions of the liver (Supplementary Fig. 9b). Furthermore, GREB1 ASO-5715 targeting mouse GREB1 mRNA tended to inhibit liver tumor formation in BYM mice and also suppressed GREB1 expression in tumor cells (Supplementary Fig. 9c, d). Thus, GREB1 ASOs may represent promising therapeutics for HB patients.

## Discussion

This study clarified the molecular mechanisms by which GREB1 promotes HB cell proliferation. GREB1 is an estrogen-induced gene product that interacts with ER, and mediates ER-dependent transcription[8,9]. In MCF7 breast cancer cells, GREB1 is induced through ligand-dependent binding of ERα to three estrogen response elements located 1.6, 9.5, and 21.2 kb upstream of the GREB1 transcription start site[31]. Here, we found the TCF4-binding site located 0.45 kb upstream of the GREB1 gene. In addition, ERα is not expressed in HepG2 cells, and ICI.182.780 did not affect GREB1 mRNA expression; in MCF7 cells, CHIR99021 did not induce GREB1 mRNA expression. Therefore, the GREB1 gene could be expressed by different mechanisms depending on cell type, and Wnt/β-catenin signal-dependent GREB1 expression may be confined to specific cells, including hepatoblasts.

The crosstalk between the Wnt and TGFβ pathways has been extensively studied, and it was shown that components of Wnt and TGFβ signaling physically interact and mutually regulate each other, resulting in the fine-tuning of the development process and homeostasis[32–34]. TGFβ usually inhibits (but sometimes stimulates) cell proliferation in a cell-type specific manner[16]. The key finding in this study is that the Wnt signal transcriptional target GREB1 binds to Smad2/3 in the nucleus and inhibits TGFβ signaling, resulting in the stimulation of cell proliferation. GREB1 is localized to the chromatin-free space in the nucleus. Although the interchromatin space contains speckles, Cajal bodies, and PML bodies[35], GREB1 was not colocalized with them but with phosphorylated Smad2/3. Since chromatin loops can expand into

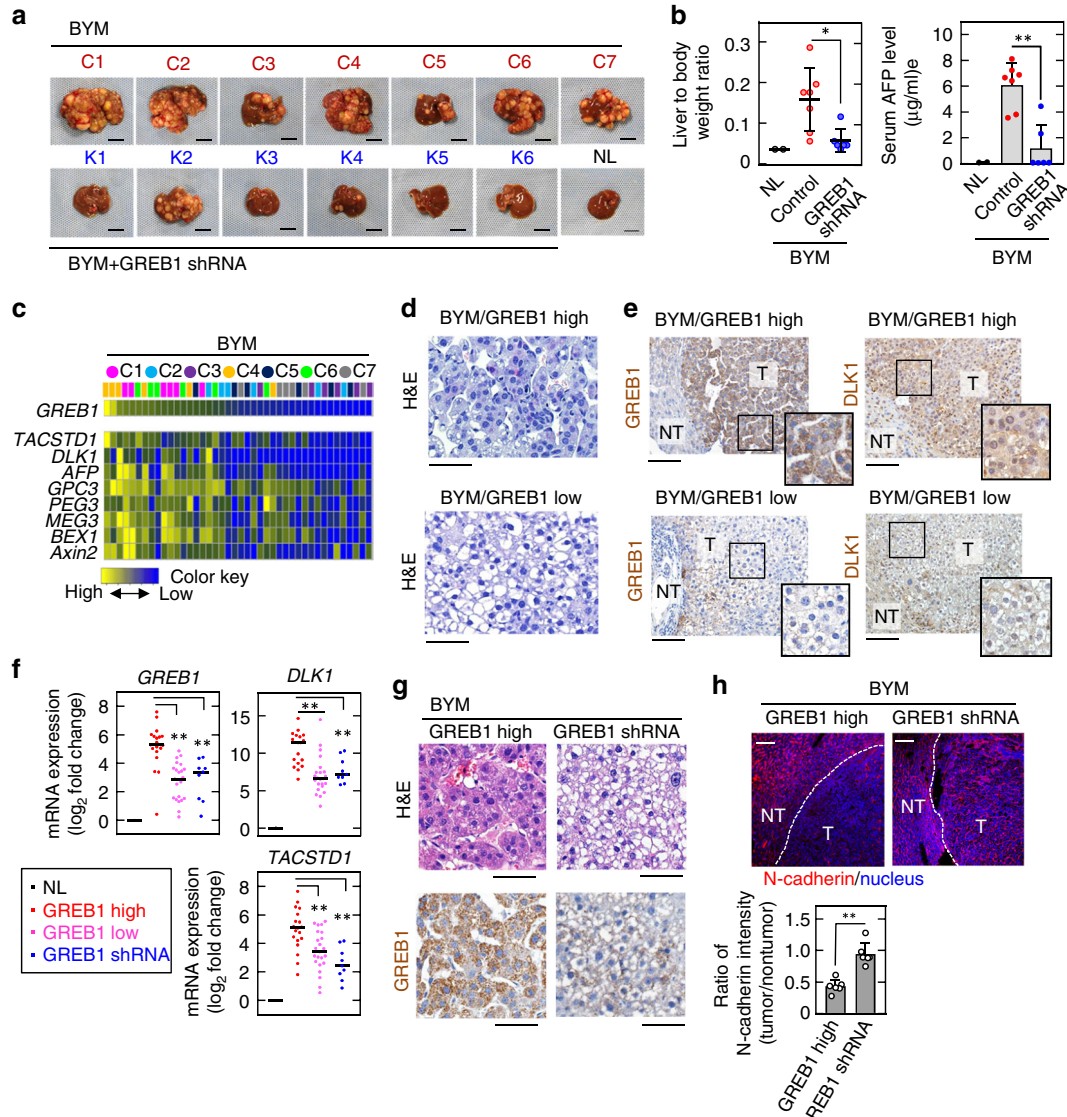

**Fig. 7** GREB1 expression is involved in HB-like tumorigenesis in vivo. **a** Images of livers from untreated mice (NL) and mice injected with ΔNβ-catenin, YAPS127A, and c-Met (BYM) with or without GREB1 shRNA at 8 weeks after hydrodynamic tail vein injection of plasmids. NL, normal liver; C, control; K, knockdown. **b** Weights of liver and whole body in untreated mice and BYM mice with or without GREB1 shRNA were measured and the results are expressed as liver weight to body weight ratio. Results are shown as dot plots with median ± SD. Serum were prepared and the amounts of AFP levels were measured by ELISA. **P < 0.01; *P < 0.05, t test. **c** Total RNA was prepared from six tumor nodules from seven BYM mice (n = 42) and real-time PCR experiments for mRNA expression of indicated genes were performed. Results standardized by min–max normalization are shown as heatmap with the highest GREB1 expression on the left. **d, e** Tissue sections of the liver from BYM/GREB1 high (C4) and BYM/GREB1 low (C3) were stained with hematoxylin and eosin (**d**) or indicated antibodies and hematoxylin (**e**). Solid squares show enlarged images. NT nontumor, T tumor. **f** Total RNA samples were prepared from NL, tumor nodules from three BYM/GREB1 high, four BYM/GREB1 low, and two BYM with GREB1 shRNA mice. Real-time PCR analyses for mRNA expression of indicated genes were performed. The obtained results are expressed as log₂ fold changes compared with NL and results are shown as dots plot with median. **P < 0.01, ANOVA and post hoc test. **g** Tissue sections of the liver isolated from the BYM mouse (C4) and the BYM mouse with GREB1 shRNA (K2) were stained with hematoxylin and eosin, or anti-GREB1 antibody and hematoxylin. **h** Tissue sections used in **g** were stained with anti-N-cadherin antibody and Hoechst33342. N-cadherin fluorescence intensity in nontumor regions and tumor lesions were measured and the results are expressed as ratio of N-cadherin expression in tumor lesions to that in nontumor regions. **P < 0.01, t test. Scale bars in **a**, 1 cm; in **c** and **g**, 50 μm; in **d** and **h**, 100 μm

the interchromatin space where transcription and splicing factors freely roam[36], GREB1 could inhibit TGFβ signaling in this space by competing with p300. These results indicate that GREB1 regulates cellular functions independently of estrogen and androgen signaling and uncovers the involvement of Wnt/β-catenin-GREB1-Smads axis in HB pathogenesis.

Several types of HB mouse models have been generated, including tissue-specific transgenic mice with expression of

c-Myc or Lin28b, and mice hydrodynamically transfected with activated β-catenin and YAP[22–24]. These murine models can phenotypically display both HCCs and HBs[25,37], however, HB mouse models that recapitulate human HB characteristics are still limited. In this study we demonstrated that forced expression of YAP with β-catenin and c-Met accelerated tumor formation with characteristics resembling human crowded fetal HB cells[24] and expression of liver progenitor marker genes compared with BY or

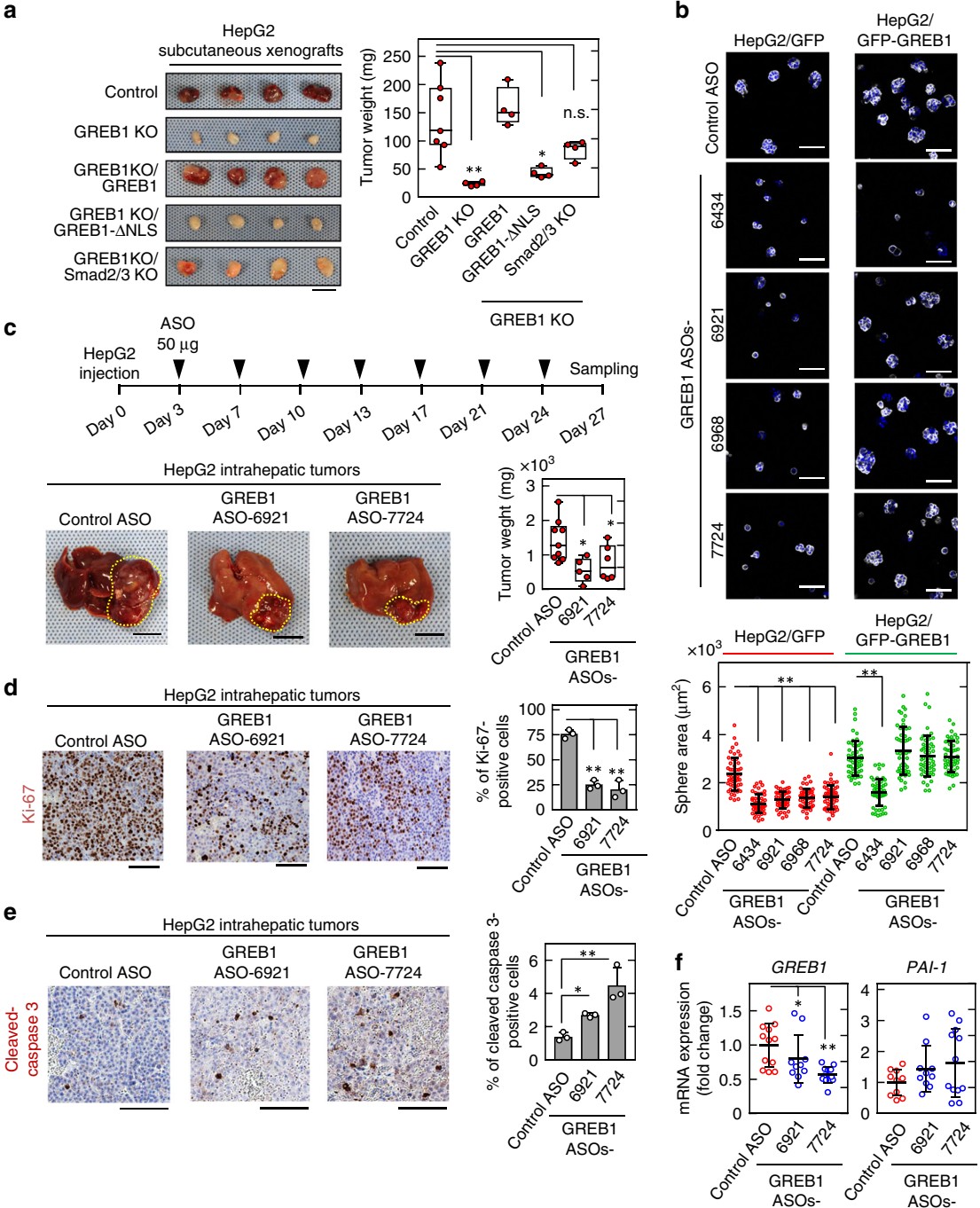

**Fig. 8** GREB1 ASOs inhibit HB cell-induced tumor formation. **a**, Control, GREB1 KO expressing the indicated proteins, or GREB1 and Smad2/3 double KO HepG2 cells were subcutaneously implanted into nude mice ($n = 4$). After 27 days, the mice were euthanized. Representative appearances of extirpated xenograft tumors are shown. Weights of the xenograft tumors were measured and the results are shown as box plots. **$P < 0.01$; *$P < 0.05$, ANOVA and post hoc test. **b** HepG2 cells transfected with control or indicated GREB1 ASOs were cultured for 4 days in 3D Matrigel. Cells were stained with phalloidin and Hoechst33342, and areas of spheres were calculated. Results are shown as scatter plots with means ± SD ($n = 50$). **$P < 0.01$, ANOVA and post hoc test. **c** HepG2 cells were implanted with Matrigel into the livers of nude mice on day 0. Starting from day 3, control ASO ($n = 9$), GREB1 ASO-6921 ($n = 5$), or GREB1 ASO-7724 ($n = 6$) was administered subcutaneously twice a week. Representative images of the liver tumors are shown. Weights of the xenograft tumors were measured and the results are shown as box plots. The yellow dotted lines indicate the outline of the tumors. *$P < 0.05$, ANOVA and post hoc test. **d**, **e** Sections from HepG2-induced liver tumors treated with control or GREB1 ASOs were stained with anti-Ki-67 (**d**) or anti-cleaved caspase 3 (**e**) antibody and hematoxylin. Ki-67 or cleaved caspase 3-positive cells were counted, and results are expressed as the percentage of positively stained cells compared with the total number of cells. **$P < 0.01$; *$P < 0.05$, ANOVA and post hoc test. **f** Real-time PCR experiments for indicated mRNA expression in liver tumors were performed. Results are shown as dot plots with median ± SD. **$P < 0.01$; *$P < 0.05$, ANOVA and post hoc test. Box-plot elements are defined as follows; median is represented by the line, the box represents the 25–75th percentiles, and the whiskers go down to the smallest value and up to the largest. Scale bars in **a** and **c**, 1 cm; in **b**, **d**, and **e**, 100 μm

BM model. However, the BYM model tumors with low expression of GREB1 contained well-differentiated cells with a clear cytoplasm and small nucleoli, which are difficult to distinguish from HCC cells. Since YAP antagonizes Wnt activity in development and human cancer[38], hyperactivation of YAP signaling was thought to suppress GREB1 expression. However, YAP did not affect GREB1 expression. Therefore, YAP could be important for rapid de-differentiation and proliferation of hepatocyte/hepatoblast in HB model. One of the reasons why c-Met promotes *GREB1* expression in liver tumors would be due to the activation of the β-catenin pathway by phosphorylation of β-catenin at Tyr654. Taken together, the BYM model would be the appropriate model for analyzing HB.

Tumors of the high GREB1 expression group highly expressed HB and undifferentiated markers compared with those of the low GREB1 expression group. ShRNA-mediated GREB1 knockdown strongly suppressed marker gene expression and HB-like liver tumorigenesis in BYM mice. From the results of GREB1 depletion in vitro and in vivo experiments, it is believed that GREB1 may represent a molecular target for treating HB. Since GREB1 is a nuclear protein, one strategy for targeting GREB1 is by using nucleic acids rather than low molecular compounds or antibodies. We synthesized GREB1 ASOs, which were flanked by AmNA with a phosphorothioate backbone. This type of ASO is stable in the blood and can be highly incorporated into liver cells in vivo[30,39,40], with the advantage of being less toxic[41]. Indeed, GREB1 ASOs suppressed HepG2-induced xenograft liver tumor formation and HB-like tumor formation in BYM model mice as well as inhibiting GREB1 expression. Thus, GREB1 ASOs may be a good candidate of molecularly targeted therapy for HB.

## Methods

**Cells and antibodies.** HepG2 cells were purchased from American Type Culture Collection (ATCC, Manassas, VA, USA). MCF7, HLE, and Huh7 cells were from Japanese Collection of Research Bioresources (JCRB, Osaka, Japan). Lenti-XTM293T (X293T) cells were from Takara Bio Inc. (Shiga, Japan). Huh6 HB cells were kindly provided by H.O. (Osaka University, Suita, Japan) in August 2015. SNU387, SNU449, and BMEL cells were kindly provided by T.K. (Osaka University, Suita, Japan) in January 2017. Authentication was provided by ATCC or JCRB.

HepG2 cells were grown in Eagle's minimum essential medium supplemented with 10% fetal bovine serum (FBS), nonessential amino acids (NEAA), and glutamax. HLE, Huh6, Huh7, SNU387, SNU449, and X293T cells were grown in Dulbecco's modified Eagle's medium (DMEM) supplemented with 10% FBS. MCF7 cells were grown in DMEM supplemented with 10% FBS, NEAA, and 1 mM of sodium pyruvate. BMEL cells were grown in DMEM/F12 supplemented with 10% FBS, 0.5 mM of sodium pyruvate, 50 ng/mL of epidermal growth factor, 30 ng/mL of insulin-like growth factor II, 10 μg/mL of insulin. All cells were stored in liquid nitrogen.

The following antibodies were used: Anti-GREB1 (#MABS62, 1:1000 for WB, 1:100 for IHC or IF), anti-phospho-HistoneH3 (ser10) (#06–570, 1:1000 for WB), anti-TCF4 (#05–511 for ChIP), and anti-acetyl-Histone H4 (#06–866 for ChIP) antibodies were from Merck Millipore (Billerica, MA, USA). Anti-HSP90 (#610419, 1:1000 for WB), anti-Smad2/3 (#610842, 1:1000 for WB), anti-β-catenin (#610154, 1:1000 for WB, 1:200 for IHC), and anti-N-cadherin (#610920, 1:100 for IF) antibodies were from BD Biosciences (San Jose, CA, USA). Anti-GREB1 (#sc-138794, 1:100 for IHC), anti-Smad4 (#sc-7966, 1:1000 for WB), anti-p300 (#sc-585, 1:1000 for WB), and anti-GFP (#sc-9996, 1:1000 for WB) antibodies were purchased from Santa Cruz Santa Cruz Biotechnology, (Dallas, TX, USA). Anti-Smad2/3 (#8685, 1:100 for IF), anti-phospho-Smad2 (Ser465/467)/Smad3 (Ser423/425) (#8828, 1:1000 for WB, 1:100 for IF), anti-cleaved caspase 3 (#9661, 1:1000 for WB, 1:100 for IHC), anti-PARP (#9532, 1:1000 for WB), anti-HistoneH3 (#9715, 1:1000 for WB), Ki-67 (#9027, 1:100 for IHC), anti-YAP1 (#14074, 1:100 for IF or IHC), anti-c-Met (#8198, 1:1000 for WB), anti-c-Myc (#5605, 1:1000 for WB), anti-TGFβ (#3711, 1:1000 for WB), and anti-Axin2 (#2151, 1:1000 for WB) antibodies were from Cell Signaling Technology (Beverly, MA, USA). Anti-β-tubulin (#T8328, 1:1000 for WB), anti-β-actin (#A5316, 1:1000 for WB), and anti-phospho-Catenin-β (pTyr654) (#SAB4504128, 1:1000 for WB) antibodies were from Sigma-Aldrich (Steinheim, Germany). Anti-GFP (#A6455, 1:2000 for WB, 1:500 for IF) antibody was from Life Technologies/Thermo Fisher Scientific (Carlsbad, CA, USA). Anti-Smad2/3 (#ab207447 for IP) antibody was from Abcam (Cambridge, UK). Anti-FLAG (#014–22383, 1:1000 for WB) antibody was from WAKO (Tokyo, Japan). Anti-DLK1 (#10636–1-AP, 1:100 for IHC) antibody was from Proteintech

Group, Inc (Chicago, IL, USA). Anti-DLK1 (#MAB8634, 1:100 for IHC) antibody was from R&D Systems, (Minneapolis, MN, USA). WB, western blotting; IHC, immunohistochemistry; IF, immunofluorescence; IP, immunoprecipitation.

**RNA-sequencing analyses.** A library from HepG2 cells transfected with control and β-catenin siRNA was prepared using a TruSeq Stranded mRNA Sample Prep Kit (Illumina, San Diego, CA, USA). Sequencing was performed on an Illumina HiSeq 2500 platform in 75-base single-end mode. CASAVA 1.8.2 software (Illumina) was used for base calling. Sequenced reads were mapped to the human reference genome sequence (hg19) using TopHat v2.0.13 in combination with Bowtie2 ver. 2.2.3 and SAMtools ver. 0.1.19. Fragments per kilobase of exon per million mapped fragments (FPKMs) were calculated using Cuffnorm version 2.2.1.

Of the 23,284 genes analyzed, 8929 genes were extracted with a normalized FPKM value greater than 3.0 in knockdown and control samples. A total of 76 genes were downregulated by more than threefold in β-catenin knockdown cells compared with control cells (P < 0.001 [Welch's *t*-test]). The binding peaks for *TCF7L2* (TCF4), a transcription-associated factor with binding profiles in HepG2 cells were downloaded as TCF7L2_HepG2_hg19_1 geneset (465 genes) from the ENCODE transcription factor binding site profiles resource (http://amp.pharm.mssm.edu/Harmonizome/gene_set/TCF7L2_HepG2_hg19_1/ENCODE + Transcription + Factor + Binding + Site + Profiles). Finally, 11 downregulated genes were identified from the ENCODE TCF4-binding site profiles (see Fig. 1a and Supplementary Table 1). Raw data files for RNA sequencing have been deposited in the NCBI Gene Expression Omnibus (GEO) database under the accession code GSE133976.

**Open source clinical data analysis.** Clinical data of HB patients were obtained and analyzed using the López–Terrada dataset (GEO ID: gse75271) from the 'R2: genomics analysis and visualization platform (http://r2.amc.nl)'. All gene expression data, P values, and r values were downloaded.

**Patients and cancer tissues.** HB tissues (n = 11) from patients who had undergone surgery at Osaka University Hospital from January 2008 to March 2015 were examined in this study. The ages of the patients ranged from 0 to 16 years (median, 3 years). Resected specimens were macroscopically examined to determine the location and size of tumors, and specimens for histology were fixed in 10% (v/v) formalin and processed for paraffin embedding. Specimens for examination were sectioned at 4 μm thickness and stained with hematoxylin and immunoperoxidase for independent evaluations by four pathologists (S.N., E.M., Y.U. and S.T). The protocol for this study was approved by the ethical review board of the Graduate School of Medicine, Osaka University, Japan (No. 13455) under Declaration of Helsinki, and were performed in accordance with the Committee guidelines and regulations. The written informed consent was obtained from all patients.

**Hydrodynamic tail vein injection and in vivo ASO treatment.** pT2BH-YAPS127A (20 μg), pT2BH-ΔN90βcatenin-Luc (20 μg), pT3-EF1a-c-MET (20 μg), and GFP2ALuc without or with mGreb1 shRNA (20 μg) and pLIVE-SB13 (8 μg) were diluted into 2.5 ml physiological saline solution and then injected into the lateral tail vein of 6–8-week-old wild-type C57BL6/N male mice (Japan SLC Inc., Hamamatsu, Japan) within 5–7 s as described[22,25]. Livers were harvested at a fixed time after Hydrodynamic Tail Vein injection (HTVi) (or the time of morbidity) to examine various parameters of tumor formation. When necessary, mouse GREB1 ASO-5715 (50 μg/body; ~2.5 mg/kg) was administered subcutaneously twice a week from day 3. Mice were euthanized at 6–7 weeks after HTVi. Tumor weights were collected and subjected to real-time PCR analyses.

**Generation of ASOs targeting for GREB1.** Phosphorothioate 15-mer ASOs containing AmNA monomers were synthesized by GeneDesign (Ibaraki, Japan)[30]. The sequences of the ASOs are listed in Supplementary Table 5. HepG2 cells were transfected with ASOs at 10 nM using RNAiMAX (Invitrogen, Carlsbad, CA, USA). Transfected cells were then used for experiments conducted at 36–48 h post transfection.

**Xenograft liver tumor formation assay and ASO treatment.** A HepG2 cell pellet (1 × 10^7 cells) was suspended in 100 μl of high concentration Matrigel (Corning) and directly injected into the livers of anesthetized 8-week-old male BALB/cAJcl-nu/nu mice (nude mice; CLEA Japan). ASO (50 μg/body; ~2.5 mg/kg) was administered subcutaneously twice a week from day 3. Mice were euthanized at 27 days after transplantation. Tumor weights were collected and subjected to histological analyses.

**Knockdown of protein expression by siRNA.** For siRNA analyses, the following target sequences were used. Randomized control, 5′-CAGTCGCGTTTGCG ACTGG-3′; *human β-catenin#1*, 5′-CCCACTAATGTCCAGCGTT-3′; *human β-catenin#2*, 5′-GCATAACCTTTCCCATCAT-3′; *human GREB1#1*, 5′-GCCATT CGTGTGCTTCCAT-3′; *human GREB1#2*, 5′-CCTCCTACAAAGCAATATT-3′; *human TGFB1:* 5′-GCGCCCATCTAGGTTATTT-3′; human c-Met: 5′-GCAT-CAGAACCAGAGGCTT-3′; *human p15,* 5′-GGAATAACCTTCCATACAT-3′;

*human YAP#1*, 5′-GACATCTTCTGGTCAGAGA-3′; *human YAP#2*, 5′-CTGG TCAGAGATACTTCTT-3′; *human TAZ#1*, 5′-ACGTTGACTTAGGAACTTT-3′;. *human TAZ#2*, 5′-AGGTACTTCCTCAATCACA-3′.

HepG2, Huh6, and MCF7 cells were transfected with the indicated siRNAs (10 nM) against genes of interest using RNAiMAX (Life Technologies/Thermo Fisher Scientific). Transfected cells were then used for experiments conducted at 48–120 h post transfection.

**Immunohistochemical analysis**. Tissue sections for immunohistochemical staining were examined using a DakoReal™EnVision™Detection System (Dako, Carpentaria, CA, USA) according to the manufacturer's recommendations[42,43]. Antigen retrieval for staining was performed using a decloaking chamber (Biocare Medical, Walnut Creek, CA, USA). Endogenous peroxidase activity was blocked with 3% H$_2$O$_2$-methanol for 15 min, and sections were then incubated with goat serum for 1 h to block nonspecific antibody-binding sites. Tissue specimens were incubated with mouse anti-GREB1 (1:100), mouse anti-β-catenin (1:100), or rabbit anti-YAP1 (1:100) antibody for 16 h at 4 °C, and binding was detected by subsequent incubation with goat anti-mouse IgG-horseradish peroxidase for 1 h. Diaminobenzidine (Dako) was used as a chromogen. Tissue sections were then counterstained with 0.1% (w/v) hematoxylin. Stained areas for β-catenin, GREB1, and YAP were classified into three categories (<5%, 5–30%, and 30–95%), and the results were considered positive when the total area of a tumor lesion showed >5% staining.

**Immunofluorescence staining**. Cells grown on glass coverslips were fixed for 10 min at room temperature in PBS containing 4% (w/v) paraformaldehyde, then permeabilized in PBS containing 0.2% (w/v) Triton X-100 and 2 mg/ml BSA for 10 min. Cells grown in 3D culture were fixed for 30 min at room temperature in PBS containing 4% (w/v) paraformaldehyde, then permeabilized and blocked in PBS containing 0.5% (w/v) Triton X-100 and 40 mg/ml BSA for 30 min. The fixed and permeabilized cells were incubated with primary antibodies for 3 h at room temperature or overnight at 4 °C, and with secondary antibodies according to the manufacture's protocol (Molecular Probes, Carlsbad, CA). Samples were viewed and analyzed using an LSM880 laser confocal microscope (Carl-Zeiss, Jena, Germany).

**Heatmap visualization of gene expression**. Each gene expression value was obtained by normalizing in *GAPDH* mRNA and was calculated as a ratio compared with normal liver. After data were standardized by min–max normalization, the heatmap was colored using Excel software (Microsoft, Redmond, WA, USA).

**Cell proliferation assay**. Cells were seeded at a density of $1.0 \times 10^4$/ml. At 12 h after plating, the medium was replaced with 10% serum medium. Cells were counted every 48 h for 8 days. Cyquant NF assays (Life Technologies/Thermo Fisher Scientific) were performed according to the manufacturer's instructions.

**Quantitative RT-PCR**. Total RNA was isolated and quantitative RT-PCR was performed[44]. Forward and reverse primers were as follows: human GAPDH, 5′-TCCTGCACCACCAACTGCTT-3′ and 5′-TGGCAGTGATGGCATGGAC-3′; human UBC, 5′-CCTGGTGCTCCGTCTTAGAG-3′ and 5′-TTTCCCAGCAAAG ATCAACC-3′; human GREB1, 5′-GTGCTCTACCGGCTCAAGTT-3′ and 5′-AC CGAGTCCACCACGTAGAT-3′; human Axin2, 5′-CTGGCTCCAGAAGATCAC AAAG-3′ and 5′-CATCCTCCCAGATCTCCTCAAA-3′; human DLK1, 5′-TGGC TTCTCAGGCAATTTCT-3′ and 5′-GGCTTGCACAGACACTCGTA-3′; human AFP, 5′-AGCTTGGTGGTGGATGAAAC-3′ and 5′-CCCTCTTCAGCAAAGCAG AC-3′; human PEG3, 5′-TCACTTCCAGCACAGCATTC-3′ and 5′-CCTCAGCC AGTGTGGGTATT-3′; human PRLR, 5′-TCATGATGGTCAATGCCACT-3′ and 5′-GCGTGAACCAACCAGTTTTT-3′; human XBP1, 5′-TCACCCCTCCAGAAC ATCTC-3′ and 5′-ACAGAGAAAGGGAGGCTGGT-3′; human PAI-1, 5′-CACAC CAGCTCCACTGAAGA-3′ and 5′-CTCCATCACAGGAGCAGACA-3′; human SNAIL2, 5′-CTTTTTCTTGCCCTCACTGC-3′ and 5′-ACAGCAGCCAGATTCC TCAT-3′; human TGFB1, 5′-CACGTGGAGCTGTACCAGAA-3′ and 5′-TGCAG TGTGTTATCCCTGCT-3′; human Met, 5′-AGAGCTGGTCCAGGCAGTGCAGC ATGTAGT-3′ and 5′-AATCTTTCATGATGATTCCCTCGGTCAGAA-3′; human p15, 5′-GACCGGGAATAACCTTCCAT-3′ and 5′-CACCAGGTCCAGTCAAGG AT-3′; human p21, 5′-ATGAAATTCACCCCCTTTCC-3 and 5′-CCCTAGGCTG TGCTCACTTC-3′; human p27, 5′-ACCCCTAGAGGGCAAGTACG-3′ and 5′-ATCAGTCTTTGGGTCCACCA-3′; human ANKRD1, 5′-ACGCCAAAGACAG AGAAGGA-3′ and 5′-TTCTGCCAGTGTAGCACCAG-3′; human Cyr61, 5′-CT CCCTGTTTTTGGAATGGA-3′ and 5′-TGGTCTTGCTGCATTTCTTG-3′; human specific β-catenin, 5′-TAGAAACAGCTCGTTGTACCGCTGGGACCT-3′ and 5′-GCACTGCCATTTTAGCTCCTTCTTGATGTAAT-3′; human specific YAP, 5′-CCACCAGTCCACCAGTGCAGCAGAATA-3′ and 5′-GCAGTCGCA TCTGTTGCTGCTGGTTGGAGT-3′; human specific Met, 5′-TGGACTCAACAG ATCTGTCTGCCTGCAATC-3′ and 5′-AATGTACTGTATTGTGTTGTCCCGTG GCCA-3′; mouse GREB1, 5′-TGATTCGGCTGACAGAAGTG-3′ and 5′-TGATG GTCTGAGGGTGTGAA-3′; mouse AFP, 5′-GCACTGTCCAAGCAAAGCTGCG CTCTCTAC-3′ and GCTGATACCAGAGTTCACAGGGCTTGCTTC-3′; mouse GPC3, 5′-GAACCATGTCTGTGCCCAAGGGTAAAGTTC-3′ and 5′-CGCTGT

GAGAGGTGGTGATCTCGTTGTCCT-3′; mouse TACSTD1, 5′-GATCTGGACC CCGGGCAGACTCTGATTTAC-3′′ and 5′-CAGCACTCAGCACGGCTAGGCA TTAAGCTC-3′; mouse DLK1, 5′-TGGCTGTGTCAATGGAGTCTGCAAGG-3′ and 5′-TGCTGGCAGGGAGAACCATTGATCACG-3′; mouse PEG3, 5′-ACTCC CTACCTTTTGGTGAGTTGCTTGCAG-3′ and 5′-CTTGGATGAAACGTTCTT GGCATAACTGG-3′; mouse BEX1, 5′-AGGCAAGGATAGGCCCAGGAGTA ATGGAGT-3′ and 5′-CTCCCCAACCCTCTGCATCAGGTCCCATCT-3′; mouse MEG3, 5′-TATCTGGACATTGAAGCTTGGAAAGCCAGT-3′ and 5′-TTCATG ACCACAGCCCATGGTATCACACAG-3′.

**Complex formation and immunoprecipitation**. HepG2 cells (100-mm diameter dish) were lysed in 400 µl of lysis buffer (10 mM Tris-HCl [pH 7.4], 140 mM NaCl, 5 mM EDTA, 1% NP40, 25 mM NaF, 20 mg/ml leupeptin, 20 mg/ml aprotinin, and 10 mM PMSF). Lysates were immunoprecipitated with anti-Smad2/3 antibody, and immunoprecipitates were probed with the indicated antibodies. To examine the complex states of GREB1, Smad3, Smad4, and Smad7, X293T cells (60-mm diameter dish) expressing HA- FLAG-GREB1, GFP-GREB1 mutants, GFP-Smad2 mutants, HA-Smad3, GFP-Smad3, GFP-Smad4, or GFP-Smad7 were lysed in 400 µl of lysis buffer. Lysates were immunoprecipitated with anti-GFP antibody and immunoprecipitates were probed with the indicated antibodies.

**In vitro GST pull-down assay**. For in vitro binding of Smad2/MH1 or Smad2/MH2 to GREB1, total cell lysates of HepG2 or Huh6 expressing GFP-GREB1 were incubated with 20 µg of GST-Smad2/MH1, or GST-Smad2/MH2 bound to glutathione-Sepharose beads for 1 h. The beads were then washed with lysis buffer three times and precipitated by the centrifugation. The precipitates were probed with anti-GREB1 or anti-GFP antibody.

**Cell labeling with ethynyl uridine (EU)**. Cells expressing GFP-Smad3 with or without HA-FLAG-GREB1 were treated with a EU at 1 mM concentration for 30 min prior to fixation and detection with the Click-iT RNA Alexa Fluor 594 Imaging Kit (Thermo Fisher Scientific,Waltham, MA, USA) according to the manufacturer's instructions.

**Xenograft subcutaneous tumor formation assay**. Male BALB/cAJcl-nu/nu mice of 5 weeks old (nude mice; CLEA Japan, Tokyo, Japan) were anesthetized with a combination of medetomidine (0.3 mg/kg body weight) and midazolam (4 mg/kg). Mice then received a dorsal subcutaneous injection of HepG2 cells ($7 \times 10^6$ cells) suspended in 100 µl of high concentration Matrigel (Corning, NY, USA). Mice were then euthanized at 28 days after transplantation, and the areas containing transplanted cells were measured, weighed and processed for immunohistochemical analyses. All protocols used for animal experiments in this study were approved by the Animal Research Committee of Osaka University, Japan (No. 26–032–048).

**Generation of GREB1-knockout cells**. The target sequence for human *GREB1*, 5′-CTTCTCGGTGTTGAAGCCGA-3′, was designed with the help of the CRISPR genome engineering resources (http://www.genome-engineering.org/crispr/). Plasmids expressing hCas9 and single-guide RNA (sgRNA) were prepared by ligating oligonucleotides into the BbsI site of pX330 (addgene#42230). The plasmid pX330 with sgRNA sequences targeting GREB1 and blasticidin resistance was introduced into HepG2 cells using Lipofectamine LTX reagent (Life Technologies/ Thermo Fisher Scientific)[45] according to the manufacturer's instructions and transfected cells were selected in medium containing 5 µg/mL blasticidin S for 2 days. Single colonies were picked, mechanically disaggregated, and replated into individual wells of 24-well plates. To establish Smad2 and Smad3 (Smad2/3) double knockout cells, plasmid pRP[CRISPR] expressing hCas9 and dual guide RNAs were designed and synthesized by VectorBuilder Inc. (Guangzhou, China). The target sequences for human Smad2 and human Smad3 were as follows: human Smad2, 5′-TATATTGCCGATTATGGCGC-3′; and human Smad3, 5′-GGAATGTCT CCCCGACGCGC-3′. The plasmid pRP[CRISPR] with dual gRNA sequences targeting Smad2/3 was introduced into HepG2 or HepG2/GREB1-knockout cells and then single colonies were picked, mechanically disaggregated, and replated into individual wells of 24-well plates.

**Chromatin immunoprecipitation (ChIP) assay**. The cells were cross-linked with 1% (v/v) formaldehyde for 20 min at room temperature. The cells were lysed with sodium dodecyl sulfate (SDS) lysis buffer (50 mM Tris-HCl (pH 8.0), 10 mM EDTA, and 0.5% (w/v) SDS)and sonicated to shear DNA to a size range between 200 and 1000 bp. Sheared chromatin samples were diluted in ChIP dilution buffer (16.7 mM Tris-HCl (pH 8.0), 167 mM NaCl, 1.2 mM EDTA, and 1.1% (w/v) Triton X-100) supplemented with protease inhibitors, and precleared with salmon sperm DNA/protein A-agarose (Millipore) and incubated for 4 h at 4 °C with 10 µg of anti-TCF4 (#05–511, Millipore), anti-β-catenin (#610154, BD Biosciences), or anti-acetyl-histone H4 antibodies (#06–866, Millipore) or negative control IgG (#kch-504–250, Diagenode, Liége, Belgium). Immunocomplexes were absorbed with salmon sperm DNA/protein A-agarose beads, and washed once with high salt buffer (20 mM Tris-HCl (pH 8.1), 500 mM NaCl, 0.1% (w/v) SDS, 1% (w/v) Triton

X-100, and 2 mM EDTA), once with LiCl buffer (10 mM Tris-HCl (pH 8.1), 0.25 M LiCl, 1 mM EDTA, 1% (w/v) deoxycholic acid, and 1% (w/v) Nonidet P-40), and three times with TE buffer (10 mM Tris-HCl (pH 8.1), and 1 mM EDTA). Immune complexes extracted in elution buffer (1% (w/v) SDS and 100 mM NaHCO3) were incubated for 4 h at 65 °C to revert DNA-protein cross-links. Then the DNA was extracted by incubation in proteinase K (final concentration of 50 μg/ml) buffer for 1 h at 45 °C. The purified DNA was used in PCR to assess the presence of target sequences[46]. The following primers were used for PCR analysis: 5′-TGGATGATA CACAGATTGCTACCAAC-3′ and 5′-GCTTGCCTTTCTCCCTGCACTAAGC-3′ for the TCF4-binding site of human GREB1; 5′-ACCGCAACGTGGTTTTCTCA CCCTATGG-3′ and 5′-AATCTTGAATCCCATAGCTGCTTGAATC -3′ for the human PAI-1 exon2.

**Plasmid construction and generation of stable transfectants**. Standard recombinant DNA techniques were used to design a plasmid harboring full length GREB1 or the various mutants. To generate NLS-fused mutants of GREB1, three copies of the NLS, derived from the SV40 T antigen[47], were fused to the N-terminus of the GFP-GREB1 mutants.

Lentiviral vectors were constructed by subcloning GFP and pEGFPC1-GREB1 into CSII-CMV-MCS-IRES2-Bsd, which was kindly provided by Dr H. Miyoshi (RIKEN BioResource Center, Ibaraki, Japan)[48]. Vectors were then transfected along with the packaging vectors, pCAG-HIV-gp, and pCMV-VSV-G-RSV-Rev, into X293T cells using the FuGENE HD transfection reagent (Roche Applied Science, Basel, Switzerland) to generate lentiviruses. To generate HepG2 cells stably expressing GFP or GFP-GREB1, $5 \times 10^4$ parental cells were plated per well in a 12-well plate and were treated with lentiviruses and 10 μg/ml polybrene, centrifuged at $1200 \times g$ for 30 min and incubated for 24 h.

pT3-EF1a-c-MET was obtained from Addgene (Cambridge, MA, USA). pLIVE-SB13 vector was a kind gift of Dr Toru Okamoto (Osaka university). pT2BH-ΔN90βcatenin-Luc was generated by in-fusion cloning of CAG promoter, Human CTNNB1 sequence lacking aa 1–90 and Luciferase into pT2BH vector (Addgene). pT2BH-YapS127A was constructed by inserting FLAG-tagged human YapS127A fragment between the EcoRI and the NotI site of the pT2BH vector. pT2BH-GFP2ALuc mouse Greb1 shRNA was constructed by inserting U6 promoter and mGreb1 shRNA fragment amplified from Mission shRNA (TRCN0000216019) (Sigma-Aldrich) from into between the PstI and the HindIII site of pT2BH-GFP2ALuc.

**Statistical analysis**. Student's t-test was used to determine if there was statistical difference between the means of two groups. Analysis of Variance (ANOVA) with Tukey, Bonferroni, or Dunnett's post hoc tests was used for comparing three or more group means. Statistical analysis was performed using Excel 2010 (Microsoft, Redmond, WA, USA) and GraphPad Prism 7 (GraphPad Software, La Jolla, CA, USA). P-values < 0.05 were considered statistically significant.

**Study approval**. The protocol for utilization of human specimens was approved by the ethical review board of the Graduate School of Medicine, Osaka University (Osaka, Japan; No. 13455). All protocols used for animal experiments in this study were approved by the Animal Research Committee of Osaka University (Osaka, Japan; No. 26-032-048).

**Reporting Summary**. Further information on research design is available in the Nature Research Reporting Summary linked to this article.

## Data availability

All data are available from the authors upon reasonable request. Raw data files for RNA sequencing have been deposited in the NCBI Gene Expression Omnibus (GEO) database under the accession code GSE133976. Full western blots are presented in source data file.

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

## Acknowledgements

We are grateful to Drs M. Yamamoto and Y. Nishikawa for discussion about histological examination of mouse liver tumor; Y. Myojin, and S. Shigeno for experimental help; A. Miyawaki, and H. Miyoshi for donating plasmids; the NGS core facility of the Genome Information Research Center at the Research Institute for Microbial Diseases of Osaka University for the support in RNA sequencing and data analysis. This work was supported by Grants-in-Aids for Scientific Research to A.K. (2016–2020) (No. 16H06374) and Grants-in-Aid for Scientific Research on Innovative Areas, "Organelle zone" to A.K. (2018–2019) (No. 18H04861) and "Cell diverse" to A.K. (2018–2019) (No. 18H05101)

from the Ministry of Education, Culture, Sports, Science and Technology of Japan, by grants from the Yasuda Memorial Foundation and the Ichiro Kanehara Foundation of the Promotion of Medical Science & Medical Care to A.K., and by Integrated Frontier Research for Medical Science Division, Institute for Open and Transdisciplinary Research Initiatives, Osaka University to A.K.

## Author contributions

Conceptualizaion: S.M., T.Y. and A.K.; Methodology: S.M., T.Y., K.S. T.K. and A.K.; Investigation: S.M., T.Y., K.S., S.N. and E.M.; Resources: Y.K., S.O. and T.K.; Writing—original draft: S.M., T.Y. and A.K.; Writing—review and editing: S.M., T.Y., S.O., T.T., E.M., H.O. and A.K.; Supervision: A.K.; Project administration: S.M. and T.Y. and A.K.; Funding acquisition: A.K.

## Additional information

**Competing interests:** The authors declare no competing interests.

