## [Peer Review File · Nature Communications]

Reviewers' comments:

Reviewer #1, Expertise: Wnt, HB (Remarks to the Author):

The study showing GREB1 as a target of the Wnt signaling playing a role in hepatoblastoma (HB) pathogenesis by Matsumoto and colleagues is a relevant study that uncovers the role of this interesting TGF signaling modulator. Overall the study is comprehensive and well executed.

However, there are some notable concerns regarding techniques and interpretation. Also, the study falls short in relevance due to small numbers of patients and use of a rather complicated model expressing three rather than 2 oncogenes. The specificity of GREB1 being regulated by -catenin versus Met versus Yap1 needs to be addressed. Major points are highlighted below:

1. Does overexpression of mutant -catenin induce expression of GREB1 in other liver tumor cell lines?
2. 11 HB cases are insufficient to make any conclusion. What was Yap status in these cases? However, the authors did assess public dataset to identify additional HB cases. However, while authors make positive correlation between GREB1 and -catenin targets (Axin-2, Dkk1 and NKD1), they don't make correlation with CTNNB1 mutations/deletions which should be done.
3. Was there any correlation between Glutamine Synthetase (GS) and GREB1 since GS may be able to also indicate -catenin mutations/deletion in HB?
4. What is the impact of -catenin siRNA on HepG2 cell proliferation or apoptosis (as has been published previously), but can that effect be rescued by GREB1 expression?
5. Since GREB1 expression is low in Huh6 cells, what is the status of -catenin and Yap in these cells and can expression of GREB1 be induced by -catenin and/or Yap expression in these cells?
6. Why would GREB1 knockdown decrease AFP and DLK1 in HepG2 cells?
7. The mechanism of GREB1 knockdown on cell death needs further mechanistic characterization.
8. The experiments of GREB1 preferentially binding to the MH2 region of Smad2/3 in the nucleus of HepG2 cells should be repeated in another HB cell line.
9. Why would GREB1 levels decrease in HepG2 cells after Met silencing? Is this because Met can regulate tyrosine phosphorylation of -catenin at residue 654 and 670? This needs to be shown. What is effect of Met silencing on Yap and -catenin in HepG2 cells? Most models of Met and delta90--catenin co-expression yield HCC. What more does Met add to Yap--catenin in HB pathogenesis? Are all tumors in this model using 3 plasmids express the 3 plasmids consistently? Was a tag used and can it be used to verify that most HB in this model are composed of the 3 plasmids and not 2? Did HCC occur in this model? How did authors distinguish the two tumors? Some data needs to be shown.
10. Likewise, A more thorough analysis of regulation of GREB1 expression by Yap1 needs to be performed as well to assure specificity of its regulation by -catenin only especially since Met appears to also regulate GREB1.
11. The data regarding heterogeneous differentiation of HB in BYM model is intriguing. The differential expression of GREB1 (like DLK1) in HB based on differentiation needs further validation especially in patient tissues.
12. The experiments with ASO were done in tumor xenograft model using just HepG2 cells. This is inadequate. More HB cells should be used. Better yet, it would be relevant to deliver ASO to BYM model or just BY model to show response to GREB1 suppression?

Reviewer #2, Expertise: Wnt, TGFb, cancer (Remarks to the Author):

GREB1 is hormone responsive gene and it is required for proliferation of hormone sensitive cancers through unknown mechanisms. In this study, Matsumoto and colleagues described a novel function of GREB1 in proliferation of hepatoblastoma (HB) tumors with beta-catenin mutation. Authors identified GREB1 as a novel beta-catenin target gene in hepatoblastoma. Knockdown of GREB1 inhibits proliferation of HB cells in vitro and in vivo. Mechanistically, GREB1 physically interacts with SMAD2/3, blocks their interaction with histone acetyl transferase p300, and inhibits TGFb signaling. Overall, this is a nice study with lots of interesting data. The proposed TGFb

inhibitory function of GREB1 is particularly interesting. If validated, it would be quite significant for people studying WNT, TGF β , and CREB1. I have following suggestions for authors to improve the manuscript.

1. The cDNA rescue experiment is not convincing. GREB1 siRNAs had significant growth inhibitory effects in HepG2 cells with ectopic expression of GREB1 (Fig. 2a). Even partial rescue cannot be claimed based on the presented data. One possible explanation is contribution of endogenous GREB1. Authors should take advantage of HepG2 CREB1 KO and CREB1 KO GFP-GREB1 cells they generated for Fig 7a, and perform rescue experiment using proliferation (Fig. 2a), western (Fig. 2d) and qRT readouts (Fig. 2c, 4b) .
2. Physical interaction between SMAD2/3-GREB1 is a very important claim. I feel that characterization of this interaction should be improved, especially when authors could not clearly define a domain of GREB1 involved in this interaction. Current experiments are based on co-IP, so it is not clear whether this is a direct interaction. Further protein interaction experiments using purified proteins would be helpful. Minimally, authors can use GST-MH1 and GST-MH2 for GST-pulldown experiments. The co-IP experiment shown in Fig. 3b lacks a negative control. Unrelated nuclear proteins such as MYC or beta-catenin can be included as control.
3. Authors showed that overexpressing of GREB1 inhibited TGF β signaling (Fig. 4g). However, the effect is modest, and there is no specificity control. Authors should repeat the experiment with transcription reporters for different signaling pathways.
4. CREB1 knockdown increased the expression of PAI1 and SNAIL2 (Fig. 4b), presumably through enhancing TGF β signaling. To confirm this, authors should examine the effect of ALK5 inhibitor on CREB1 siRNA-induced expression of PA-1 and SNAIL2.
5. The quality of Fig. 4e can be improved. It seems that loadings for p300 and GFP-SMAD2/C are lower.
6. Fig 4h is a key figure. However, knockdown of CREB1 protein was not measured. It is possible that knockdown efficiencies varied among different experiment conditions. It is also not clear whether the effect of TGF β 1 siRNA is on target. Confirming this finding using alternative rescue strategies (ALK5 inhibitor, SMAD4 KO) would be extremely helpful.
7. I understand the manuscript is HB-focused. However, considering many studies of CREB1 in hormone sensitive tumors, it would be great to see whether tamoxifen affects TGF β signaling in MCF7 cells with and without CREB1 knockdown. It would give us a sense how general the proposed regulation is.
8. Data shown in Fig. 7b are impressive. If CREB KO inhibits cell proliferation through enhancing TGF β signaling, knockout of SMAD4 in HepG2 CREB KO cells should rescue their growth in vivo. This seems to be an easy but critical experiment.
9. Fig. 6. The experiment was done with or without mGREB1 shRNA in hydrodynamic injection. The better control would be non-targeting shRNA. It would be difficult to repeat the mouse experiment. With this in mind, clean rescue experiments in HepG2 would be important.

Our responses to the Reviewers' comments for NCOMMS-18-31601-T

Title: GREB1, a novel target of Wnt signaling, promotes development of hepatoblastoma by suppressing TGF β signaling

Authors: Shinji Matsumoto, Taku Yamamichi, koei Shinzawa, Yuuya Kasahara, Satoshi Nojima, Takahiro Kodama, Satoshi Obika, Tetsuo Takehara, Eiichi Morii, Hiroomi Okuyama, and Akira Kikuchi.

The reviewers' original comments are written in *italics* and our responses are in plain text. Our corrections in the text are underlined.

Reviewer 1

1. Does overexpression of mutant β -catenin induce expression of GREB1 in other liver tumor cell lines?

In addition to HLE cells shown in Supplementary Fig. 1c (original manuscript), GREB1 expression in other adult liver hepatocellular carcinoma (HCC) cells, such as SNU387, SNU449, and Huh7 cells, were examined in the presence or absence of CHIR99021, which activates the Wnt/ β -catenin pathway more efficiently than overexpression of a constitutively active mutant of β -catenin. Expression of *Axin2* mRNA, a well-known direct target gene of the Wnt/ β -catenin pathway, but not that of *GREB1* mRNA, was induced by CHIR99021 in HCC cells. However, *GREB1* mRNA was expressed by CHIR99021 in a bipotential mouse embryonic liver (BMEL) cell line. In addition, GREB1 expression was downregulated by β -catenin knockdown in Huh6 hepatoblastoma cells harboring active mutation in β -catenin. GREB1 expression of Huh6 cells is lower than HepG2, but higher than HCC cells. These results suggest that GREB1 is a downstream target gene for Wnt/ β -catenin signaling specifically in HB and immature liver progenitor cells. The results are shown in Supplementary Fig. 1b, c, d, and f, and described in the text (page 5, line 4 from the bottom through page 6, line 2; page 6, lines 4 and 5; page 6, lines 7 and 8).

*2. 11 HB cases are insufficient to make any conclusion. What was Yap status in these cases? However, the authors did assess public dataset to identify additional HB cases. However, while authors make positive correlation between GREB1 and β -catenin targets (*Axin-2*, *Dkk1* and *NKD1*), they don't make correlation with *CTNNB1* mutations/deletions which should be done..*

According to the reviewer's suggestion, 11 HB tissues were stained with anti-YAP antibody. YAP was detected in the cytoplasm and/or nucleus of the tumor lesions of 9 (81.8%) out of the 11 cases, and all 9 cases were also positive for GREB1 and β -catenin. The results are shown in Fig. 6a and described in the text (page 13, lines 8 through 10 from the bottom). However, as responded to comment (5), GREB1 expression was not affected by YAP/TAZ knockdown in HepG2 and Huh6 cells.

A public dataset of HB patients (GEO ID: gse75271) used in this study provides data of targeted sequencing of exons 3 and 4 of the *CTNNB1* gene. We examined the correlation between *GREB1* expression and *CTNNB1* mutations/deletions in HB tissues and found that *GREB1* expression is unchanged in HB cases with wild-type *CTNNB1* and those with *CTNNB1* mutations and deletions. The expression of Wnt/ β -catenin signaling target genes such as *GS* and *LGR5* was also unchanged in HB cases regardless of the presence or absence of *CTNNB1* mutations. These results suggest that *GREB1* expression in HB is correlated with β -catenin signaling activity but not with mutations/deletions status in exons 3 and 4 of *CTNNB1*. Other *CTNNB1* mutations or *CTNNB1* mutations-independent activation of β -catenin signaling could occur in these cases. The results are shown in Supplementary Fig. 1j and described in the text (page 7, lines 3 through 10).

3. Was there any correlation between Glutamine Synthetase (GS) and GREB1 since GS may be able to also indicate β -catenin mutations/deletion in HB?

Public dataset revealed that there is a significant positive correlation between expression levels of *GREB1* and *GS*. The results are shown in Fig. 1g and described in the text (page 6, lines 2 and 3 from the bottom).

4. What is the impact of β -catenin siRNA on HepG2 cell proliferation or apoptosis (as has been published previously), but can that effect be rescued by GREB1 expression?

Consistent with the previous report, β -catenin siRNA indeed resulted in reduction in HepG2 cells proliferation. We found that exogenous *GREB1* expression partially rescues the anti-proliferative phenotype induced by β -catenin knockdown. These results suggest that *GREB1* is one of the mediators of cell proliferation downstream of β -catenin signaling in HB cells and other factors are also involved in the regulation. The results are shown in Supplementary Fig. 2a and described in the text (page 7, lines 11 through 13 from the bottom).

5. Since *GREB1* expression is low in Huh6 cells, what is the status of β -catenin and Yap in these cells and can expression of *GREB1* be induced by β -catenin and/or Yap expression in these cells?

Huh6 cells harbors G34V somatic activating mutation in *CTNBB1* gene. β -Catenin knockdown significantly suppressed *GREB1* expression in Huh6 cells. Immunostaining analysis revealed that nuclear YAP accumulation was observed in Huh6 cells more significantly than HepG2 cells. YAP/TAZ knockdown in Huh6 cells did not affect *GREB1* expression. Treatment of Huh6 cells with CHIR99021 and/or XMU-MP-1, which inhibits Mst1/2 kinase activities to activate YAP, did not affect *GREB1* expression. These results suggest that β -catenin but not YAP is involved in *GREB1* expression in Huh6 cells. The results are shown in Supplementary Fig. 1c and Fig. 7a-c, and described in the text (page 5, line 4 from the bottom through page 6 line 2; page 14, lines 4 through 12).

6. Why would *GREB1* knockdown decrease *AFP* and *DLK1* in HepG2 cells?

It has been reported that TGF β signaling suppresses fetal liver progenitor gene *AFP* expression in rat fetal hepatocytes and HCC cells. Indeed, TGFBR^{T204D} expression in HepG2 cells decreased *AFP* and *DLK1* expression, and Smad2/3 knockout rescued *GREB1* knockdown-induced decrease in *AFP* gene expression. Therefore, *GREB1* knockdown increases TGF β signaling, thereby inhibiting *AFP* expression in HepG2 cells. However, since Smad2/3 knockout did not rescue *DLK1* gene suppression by induced by *GREB1* knockdown, the mechanism of TGF- β signal-mediated regulation of *DLK1* expression is elusive. The results are shown in Supplementary Fig. 4f-h and described in the text (page 11, lines 7 through 14).

7. The mechanism of *GREB1* knockdown on cell death needs further mechanistic characterization.

We examined expression of cytostatic genes, including *p21*, *p15*, and *p27*, which are induced by TGF β signaling. Among them, *p15* was dramatically increased by *GREB1* knockdown in HepG2 cells and the phenotype was canceled by Smad2/3 knockout. *p15* knockdown in HepG2 cells restored *GREB1* depletion-induced growth suppression and cell death, suggesting that *p15* is a key mediator of both growth inhibition and cell cycle arrest, followed by cell death. The results are shown in Supplementary Fig. 5e-h and described in the text (page 10, line 10; page 11, line 3 from the bottom through page 12, line 4).

8. The experiments of *GREB1* preferentially binding to the MH2 region of Smad2/3 in the nucleus of

HepG2 cells should be repeated in another HB cell line.

As described, Huh6 cells are an HB cell line which expresses GREB1 endogenously although the expression levels are low as compared with HepG2 cells. GFP-GREB1 expressed in Huh6 cells was coimmunoprecipitated with endogenous Smad2/3, and recombinant GST-Smad2/MH2 interacted with GFP-GREB1 expressed in Huh6 cells, supporting that GREB1 directly interacts with the MH2 domain of Smad2/3 in HB cells. The results are shown in Supplementary Fig. 3b,d and described in the text (page 9, line 10 and 11; lines 7 through 11 from the bottom).

9. Why would GREB1 levels decrease in HepG2 cells after Met silencing? Is this because Met can regulate tyrosine phosphorylation of β -catenin at residue 654 and 670? This needs to be shown. What is effect of Met silencing on Yap and β -catenin in HepG2 cells? Most models of Met and delta90- β -catenin co-expression yield HCC. What more does Met add to Yap- β -catenin in HB pathogenesis? Are all tumors in this model using 3 plasmids express the 3 plasmids consistently? Was a tag used and can it be used to verify that most HB in this model are composed of the 3 plasmids and not 2? Did HCC occur in this model? How did authors distinguish the two tumors? Some data needs to be shown.

We examined the effects of Met knockdown on β -catenin and YAP signaling. Met knockdown decreased *Axin2* and *GREB1* expression, but rather increase expression of *ANKRD1* and *Cyr61*, downstream genes of YAP, in HepG2 cells. Expression of β -catenin and YAP mRNAs was not changed by c-Met knockdown. As the reviewer mentioned, HGF/Met signaling has been shown to phosphorylate β -catenin at tyrosine 654 and to induce β -catenin nuclear translocation and activation. Indeed, Met knockdown decreased the phosphorylation of β -catenin at tyrosine 654 as well as expression of *Axin2* and *GREB1*. These results imply that Met silencing suppresses β -catenin signaling by inhibiting the tyrosine phosphorylation of β -catenin, thereby decreasing *GREB1* expression. The results are shown in Supplementary Fig. 7d,e, and described in the text (page 14, lines 7 through 13 from the bottom; page 19, lines 12 through 14).

Our preliminary experiments to establish the hepatoblastoma model demonstrated that hydrodynamic genes delivery of the combination of YAPS127A and Δ N β -catenin (BY model) induces small solid liver nodules but the tumors little express *GREB1* and *DLK1*. The combination of c-Met and Δ N β -catenin genes (BM model) induced liver nodules expressing *GREB1* and *DLK1* to the small extent. In contrast, the combination of Δ N β -catenin, YAPS127A, and Met (BYM model) induced larger liver tumors than BY or BM model and the tumor nodules highly expressed both *GREB1* and *DLK1*. Tumor nodules of BYM model expressed *GREB1* mRNA and another HB

marker *TACSTD1* mRNA higher than that of BY or BM model. Therefore, it seems to us that BYM model is appropriate for the *in vivo* analysis of GREB1 functions in HB. The results are shown in Fig. 6b,c and described in the text (page 13, line 6 from the bottom through page 14, line 3).

According to the reviewer's comment, we examined the levels of exogenous $\Delta N\beta$ -catenin, *YapS127A*, and *Met* mRNAs in individual BYM-tumor nodules by the real-time PCR since these exogenous genes were derived from human and could be detected using human-specific primer. Real-time PCR analyses revealed that BYM-tumor nodules express all three exogenous genes although the expression levels were variable. These results suggest that the features of tumors induced by BYM differ from those induced by BY or BM in terms of tumor growth and gene expression patterns. The results are shown in Supplementary Fig. 8a and described in the text (page 14, lines 2 through 4 from the bottom).

However, BYM model tumors with low expression of GREB1 contained well-differentiated cells with a clear cytoplasm and small nucleoli, which are difficult to distinguish from HCC cells. These statements are described in the text (page 19, lines 4 through 9).

10. Likewise, A more thorough analysis of regulation of GREB1 expression by Yap1 needs to be performed as well to assure specificity of its regulation by β -catenin only especially since Met appears to also regulate GREB1.

We investigated the involvement of YAP and TAZ (YAP/TAZ) in the regulation of GREB1 expression. YAP/TAZ knockdown had no effect on GREB1 expression in HepG2 and Huh6 cells, suggesting that YAP/TAZ is necessary for promoting HB tumorigenesis but is not essential for GREB1 expression. The results are shown in Supplementary Fig. 7b and described in the text (page 14, lines 7 and 8; page 19, lines 9 through 12).

11. The data regarding heterogeneous differentiation of HB in BYM model is intriguing. The differential expression of GREB1 (like DLK1) in HB based on differentiation needs further validation especially in patient tissues.

According to the reviewer's suggestions, we investigated the heterogeneous expression of GREB1 within the same patient tissues. There were two types of tumors in HB tissues. One showed solid structures with unpolarized cells, and the other had tubular structures with polarized cells. GREB1 was specifically expressed in the former tumors. These results are consistent with phenotypes of HepG2 cells that GREB1 knockdown induces epithelial polarization and differentiation (see Fig. 2b).

The results are shown in Supplementary Fig. 1h and described in the text (page 6, lines 9 through 11 from the bottom).

12. The experiments with ASO were done in tumor xenograft model using just HepG2 cells. This is inadequate. More HB cells should be used. Better yet, it would be relevant to deliver ASO to BYM model or just BY model to show response to GREB1 suppression?

As described, Huh6 cells are another HB cell line. Unfortunately, it was hard to form orthotopic xenograft tumors by Huh6 in immunodeficient mice. Therefore, we examined effects of GREB1 ASO on the BYM model. First we designed and synthesized three AmNA-modified ASOs targeting mouse *GREB1* mRNA because our original ASOs target human *GREB1* mRNA. mGREB1 ASO-5715 was administered subcutaneously twice a week from day 3 post hydrodynamic tail vein injection. GREB1 ASO-5715 inhibited liver tumor formation in BMY mice and suppressed GREB1 expression. The results are shown in Supplementary Fig. 9c,d and described in the text (page 17, lines 2 through 4 from the bottom).

Reviewer 2

(1) The cDNA rescue experiment is not convincing. GREB1 siRNAs had significant growth inhibitory effects in HepG2 cells with ectopic expression of GREB1 (Fig. 2a). Even partial rescue cannot be claimed based on the presented data. One possible explanation is contribution of endogenous GREB1. Authors should take advantage of HepG2 CREB1 KO and CREB1 KO GFP-GREB1 cells they generated for Fig 7a, and perform rescue experiment using proliferation (Fig. 2a), western (Fig. 2d), and qRT readouts (Fig. 2c, 4b).

Although the reviewer claimed that ectopic expression of GREB1 partially rescued the growth inhibitory effect by GREB1 siRNA, our results in Fig. 2a showed that GREB1 expression fully rescued growth activity of HepG2 cells to the control level. The reviewer may have compared rescue HepG2 cells with HepG2 cells expressing GREB1.

Since the reviewer's comment is important, we performed additional experiments using HepG2 GREB1 knockout cells. GREB1 KO in HepG2 cells significantly reduced cell proliferation and GFP-GREB1 expression completely rescued proliferation defects. Consistent with the phenotypes of GREB1 knockdown, the levels of *AFP* and *DLK1* mRNAs were decreased. The *PAI-1* mRNA was increased and the levels of cyclin A and phosphorylated H3 were decreased in GREB1 KO HepG2 cells. These phenotypes were rescued in GREB1 KO HepG2 cells expressing GFP-GREB1. The

results are shown in Supplementary Fig. 2g-j and Supplementary Fig. 4b, and described in the text (page 8, lines 11 through 14; page 10, lines 13 and 14).

(2) Physical interaction between SMAD2/3-GREB1 is a very important claim. I feel that characterization of this interaction should be improved, especially when authors could not clearly define a domain of GREB1 involved in this interaction. Current experiments are based on co-IP, so it is not clear whether this is a direct interaction. Further protein interaction experiments using purified proteins would be helpful. Minimally, authors can use GST-MH1 and GST-MH2 for GST-pulldown experiments. The co-IP experiment shown in Fig. 3b lacks a negative control. Unrelated nuclear proteins such as MYC or beta-catenin can be included as control.

Since GREB1 consists of 1954 amino acids, it was hard to purify GREB1 for the use of biochemical experiments. Furthermore, deletion mutants of GREB1 showed less binding activity to Smad2 than full-length GREB1 as shown in Fig. 3g, suggesting that three-dimensional structures of GREB1 is necessary for its binding to Smad2. Therefore, we prepared GST-Smad2/MH1 and GST-Smad2/MH2 recombinant proteins, and *in vitro* pull down assays were performed. GST-Smad2/MH2 could precipitate endogenous GREB1 in HepG2 and Huh6 cell lysates but GST-Smad2/MH1 did not, suggesting that the interaction of Smad2/3 with GREB1 is direct. The results are shown in Fig. 3e and Supplementary Fig. 3d and described in the text (page 9, lines 7 through 11 from the bottom).

We confirmed that nuclear proteins, β -catenin and c-Myc, did not interact with GFP-Smad3, 4, or 7 in X293T cells. The results are shown in Supplementary Fig. 3c and described in the text (page 9, lines 11 and 12).

(3) Authors showed that overexpressing of GREB1 inhibited TGF β signaling (Fig. 4g). However, the effect is modest, and there is no specificity control. Authors should repeat the experiment with transcription reporters for different signaling pathways.

We repeated the experiment of original Fig. 4g using HepG2 cells and added new results showing that GREB1 overexpression significantly inhibited TGFBR^{T204D}-induced upregulation of *SNAIL2* and *p15* mRNAs but did not affect that of *Axin2* mRNA, which is a target gene of Wnt signaling. The results are shown in Fig. 4h and described in the text (page 11, lines 4 and 5; lines 6 and 7).

(4) CREB1 knockdown increased expression of PAI-1 and SNAIL2 (Fig. 4b), presumably through enhancing TGF β signaling. To confirm this, authors should examine the effect of ALK5 inhibitor on CREB1 siRNA-induced expression of PAI-1 and SNAIL2.

According to the reviewer's suggestion, we examined the dependency of GREB1 siRNA-induced PAI-1 and SNAIL2 expressions on TGF β signaling. Treatment of ALK5 inhibitor counteracted upregulation of PAI-1 and SNAIL2 mRNAs induced by GREB1 knockdown. The results are shown in Fig. 4d and described in the text (page 10, lines 10 and 11 from the bottom).

(5) The quality of Fig. 4e can be improved. It seems that loadings for p300 and GFP-SMAD2/C are lower.

We repeated the experiment of original Fig. 4e and the improved Western blots are shown as Fig. 4f.

(6) Fig 4h is a key figure. However, knockdown of CREB1 protein was not measured. It is possible that knockdown efficiencies varied among different experiment conditions. It is also not clear whether the effect of TGF β 1 siRNA is on target. Confirming this finding using alternative rescue strategies (ALK5 inhibitor, SMAD4 KO) would be extremely helpful.

According to the reviewer's comment, knockdown efficiencies of GREB1 and TGF β 1 were examined in Western blotting. The results are shown in Supplementary Fig. 5c.

To clearly show the involvement of TGF β signaling in GREB1 knockdown phenotypes of original Fig. 4h, we knocked out Smad2 and 3 (Smad2/3) in HepG2 cells instead of Smad4 since Smad4 is a common downstream mediator for both the TGF β and BMP signaling pathways. Smad2/3 KO inhibited the decrease in cell proliferation induced by GREB1 knockdown, supporting the idea that GREB1 knockdown phenotype is mediated by the increase of TGF β signaling. The results are shown in Fig. 4j and described in the text (page 12, lines 3 and 4).

(7) I understand the manuscript is HB-focused. However, considering many studies of CREB1 in hormone sensitive tumors, it would be great to see whether tamoxifen affects TGF β signaling in MCF7 cells with and without CREB1 knockdown. It would give us a sense how general the proposed regulation is.

We thank to the reviewer for the nice advice. In this experiment we used ICI.182.780, a specific inhibitor for the ERs, instead of tamoxifen, since ICI.182.780 was reported to be more effective as an

ER inhibitor than tamoxifen and tamoxifen was water-insoluble and unsuitable for *in vitro* experiments. Treatment of ICI.182.780 or GREB1 knockdown increased *PAI-1* mRNA expression in MCF7 breast cancer cells. In addition, immunoprecipitation assay revealed that GREB1 interacts with Smad2/3 in MCF7 cells, suggesting that GREB1 functions as a negative regulator not only in HB but also in breast cancer cells. However, the biological relevance of GREB1 and TGF β signaling are currently unknown in breast cancer, and we would like to clarify this point as a next project. The results are shown in Supplementary Fig. 5i-k and described in the text (page 12, lines 6 through 9).

(8) Data shown in Fig. 7b are impressive. If CREB KO inhibits cell proliferation through enhancing TGF β signaling, knockout of SMAD4 in HepG2 CREB KO cells should rescue their growth in vivo. This seems to be an easy but critical experiment.

Since SMAD 4 is a common SMAD which is involved in TGF β and BMP signaling, we knocked out Smad2 and 3 (Smad2/3) in GREB1 KO HepG2 cells. It took considerable time to establish Smad2/3 and GREB1 triple KO HepG2 cells because of the slow growth rate of GREB1 KO cells and the low frequency of Smad2/3 double KO by CRISPR-Cas9 system.

In vitro proliferation assay revealed that Smad2/3 KO partially rescued the suppression of cell proliferation induced by GREB1 KO. Consistently, Smad2/3 KO rescued impaired xenograft tumor formation by GREB1 KO. Thus, GREB1 could abrogate TGF β signal-dependent inhibition of cell proliferation *in vivo*. The results are shown in Fig. 8a and Supplementary Fig. 8e,f and described in the text (page 16, lines 9 through 12).

In this experiment we noticed that tumor nodules from HepG2/GREB1 KO/Smad2/3 KO tended to show less vascularized and lighter color compared with those from control cells. These incomplete rescue phenotypes might be caused by the complete loss of TGF β -Smad pathway that has oncogenic functions in the late stage of cancer progression, or DNA damage induced by repeated genome editing by CRISPR/Cas9 system. The interpretation of the results needs to be considered carefully. We do not further perform additional experiments in this issue, but continue to design the experiment to clarify the functional relationship between GREB and TGF β -signaling in different stages of HB tumor formation.

(9) Fig. 6. The experiment was done with or without mGREB1 shRNA in hydrodynamic injection. The better control would be non-targeting shRNA. It would be difficult to repeat the mouse experiment. With this in mind, clean rescue experiments in HepG2 would be important.

We appreciate the reviewer's reasonable advice. As mentioned by the reviewer, it is difficult to repeat all of the mouse experiments in Fig. 6 using non-targeting shRNA as control. It is also hard to do rescue experiments with HepG2 cells because shRNA used in Fig. 6 is specific for mouse GREB1. Therefore, we performed clean rescue experiments using GREB1 knockout HepG2 cells. The results are shown in Supplementary Fig. 2g-j and in the text (page 8, lines 11 through 14).

In addition, we rescued GREB1 knockdown phenotypes by Smad2/3 knockout as shown in Fig. 4j and Supplementary Figs. 4g and 5f, to show that loss of function phenotypes of GREB1 are mediated by the inhibition of TGF β specific signaling (page 11, lines 9 and 10; lines 1 through 3 from the bottom; page 12, lines 3 and 4).

REVIEWERS' COMMENTS:

Reviewer #1 (Remarks to the Author):

The authors should be commended for responding to all comments raised by this reviewer in an optimal and comprehensive manner. There are no additional comments.

Satdarshan P Monga, MD

Reviewer #2 (Remarks to the Author):

Authors have sufficiently addressed my points.